# Passive processing of active nodal seismic data: Estimation of $V_P/V_S$ - ratios to characterize structure and hydrology of an alpine valley infill

Michael Behm[1], Feng Cheng[2], Anna Patterson[1], Gerilyn S. Soreghan[1]

[1]School of Geology and Geophysics, University of Oklahoma, Norman, OK, United States

[2]Lawrence Berkeley National Laboratory, Berkeley, CA, United States

*Correspondence to*: Michael Behm (Michael.Behm@ou.edu)

**Abstract.** The advent of cable-free nodal arrays for conventional seismic reflection and refraction experiments is changing the acquisition style for active source surveys. Instead of triggering short recording windows for each shot, the nodes are continuously recording over the entire acquisition period from the first to the last shot. The main benefit is a significant increase

in geometrical and logistical flexibility. As a by-product, a significant amount of continuous data might also be collected. These data can be analysed with passive seismic methods and therefore offer the possibility to complement subsurface characterization at marginal additional cost. We present data and results from a 2.4 km long active source profile which has been recently acquired in Western Colorado (US) to characterize the structure and sedimentary infill of an over-deepened alpine valley. We show how the 'leftover' passive data from the active source acquisition can be processed towards a shear

wave velocity model with seismic interferometry. The shear wave velocity model supports the structural interpretation of the active P-wave data, and the P-to-S-wave velocity ratio provides new insights into the nature and hydrological properties of the sedimentary infill. We discuss the benefits and limitations of our workflow and conclude with recommendations for acquisition and processing of similar data sets.

## 1 Introduction

Seismic nodal acquisition systems ('nodes' thereafter) were introduced to the active source exploration community within the last decade with the promise of geometrical flexibility and a more efficient production, especially in rugged terrain (Freed, 2008; Dean et al., 2013). Nowadays several outfitters provide instruments for a wide range of applications with a focus on the energy industry (Dean et al., 2018), but nodal acquisition is also becoming widespread in the academic community (Karplus and Schmandt, 2018). Nodes differ from conventional cable-based systems in several aspects. During recording, each node is

an autonomous data logger and recorder without required physical or non-physical connection to a central processing system. They are designed to record continuously throughout the entire acquisition period, which might last from days to months. In that regard, the acquired data can be considered as passive data which automatically include the shot windows from the active sources. For any active seismic exploration study, the shot windows are considered as the complete data set to represent the subsurface. In the case of continuous nodal acquisition, a significant amount of additional data is recorded outside the shot

windows. The lack of well-defined sources outside the active shooting times does not mean that these periods are seismically quiet. The ambient noise spectrum covers a wide frequency range and stems from diverse natural and anthropogenic processes

(McNamara and Bulland, 2004; Riahi and Gerstoft, 2015). The location and timing of specific events within this noise spectrum might be known with some degree of uncertainty (e.g. local, regional, and global seismicity), thus inviting classical active processing methods like travel time tomography to derive local velocity models (Kissling, 1988; Byriol et al., 2013) or different forms of receiver-side reflectivity mapping (Ruigrok et al., 2010; Behm and Shekar, 2014, Behm, 2018). For the more general

case of unknown locations and timing of the sources in the ambient noise spectrum (e.g. traffic noise, industrial activities) the seismic interferometry method (Snieder, 2004; Wapenaar, 2004; Schuster, 2010) has become a staple for subsurface modelling and interpretation. In particular, the extraction of surface waves travelling between receivers in locally deployed arrays can be feasible for even relatively short time spans of ambient noise. (e.g. Nakata et al., 2011; Behm et al., 2014; Cheng et al. 2016). The reconstructed surface waves are mostly used to image the local shear-wave velocity structure (e.g. Picozzi et al., 2009;

Hannemann et al., 2014) or for interpretation of temporal changes in the subsurface (e.g. Planes et al., 2015; Riahi et al., 2013). Applied to active data, interferometric surface wave removal (Halliday et al., 2007, 2010) can successfully model and mitigate unwanted Rayleigh-wave energy in shot gathers. Although body waves are much more challenging to extract from surface recordings of ambient noise (Forghani and Snieder, 2010), the availability of many stations can facilitate signal processing routines to focus on the extraction of diving waves (Nakata et al. 2015) and reflected waves (Draganov et al., 2009) as well.

Body waves caused by surface noise sources are also more likely to be detected in boreholes (Behm, 2017; Zhuo and Paulssen, 2017) or inside mines (Olivier et al., 2015).

Processing of passive data provides complementary information when compared to the active data. E.g. surface wave inversion obtained from interferometry results in shear wave velocity models, and travel time tomography using local or regional seismicity can increase the investigation depth. Strobbia et al. (2011) applied a workflow to isolate and invert Rayleigh waves

from a dense active source 3D acquisition, and in a later step used the obtained near-surface shear wave velocity model to improve the filtering of Rayleigh wave energy for reflection processing. Most of the passive processing schemes provide subsurface models with significantly lower lateral resolution than models obtained from active data. However, robust low-resolution information can be beneficial when implemented into initial models for full waveform inversion (Sirgue and Pratt, 2004; Denes et al., 2009).

From a geologic point of view, our study focuses on the structure and sedimentary infill of a presumably over-deepened alpine valley in Western Colorado (US). Alpine valleys are of interest for geophysical investigation because of their significance for landform evolution (e.g. incision rates, timing and effects of glacial overprinting; de Franco et al., 2009; Pomper et al., 2017) and their potential for harbouring significant groundwater resources (e.g. Pugin et al., 2014). Brueckl et al. (2010) provide an overview of geophysical exploration of glacially over-deepened valleys in the Austrian Alps of Europe. They report P-wave

velocities and densities for Pleistocene sedimentary infill, and in all cases, they find a deeper sedimentary layer ("old valley fill") above the bedrock with higher P-wave velocities. Bleibinhaus and Hilberg (2012) investigate one of the largest over-deepened valleys in the European Alps with seismic and electrical resistivity methods. Based on increased seismic velocities and increased resistivity, they interpret an aquifer in the shallow part of the sediments.

In our study, we present data and results from a local 2D reflection line acquired for imaging Unaweep Canyon on the northeastern Colorado Plateau. Nodal instruments recorded continuously for the duration of 2.5 days and captured shots from an active source as well as traffic-induced ambient noise. We apply seismic interferometry to the continuous data to extract dispersive surface waves, which in turn are inverted for a 2D shear-wave velocity model of the valley structure. This model complements the results from active source processing, and the joint interpretation of the active P-wave velocity and passive S-wave velocity models allows for new insights on the nature and hydrologic properties of the sedimentary valley infill.

## 2 Area and Geology

The area of investigation (Fig. 1) is the western part of the NE-SW-trending Unaweep Canyon of the Uncompahgre Plateau, western Colorado. This plateau is a large Cenozoic uplift on the northeastern Colorado Plateau and had a late Paleozoic existence as the "Uncompahgre uplift" – one of several basement-cored uplifts with paired basins that formed as part of the Ancestral Rocky Mountains (ARM) of western equatorial Pangaea (Kluth and Coney, 1981). Unaweep Canyon is an enigmatic landform since the modern drainage divide occurs in the middle of the canyon, such that it hosts two creeks that drain to both of its mouths. The canyon is deep (>400 m in inner Precambrian-hosted gorge), wide (locally >6000 m, 800 m in inner gorge), and incised into Mesozoic strata and Precambrian crystalline basement. The canyon bottom hosts sedimentary fill of Quaternary and possibly older age, at least 330 m thick in some regions (Soreghan et al., 2007).

Most suggest that the canyon was formed by the ancestral Gunnison River, and/or Colorado River in the late Cenozoic and later abandoned (e.g., Cater 1966; Sinnock, 1981; Lohman, 1961; Hood, 2011; Aslan et al., 2014). Many attributes of the canyon, however, are inconsistent with a purely fluvial origin, such as the lack of dendritic tributary systems, and apparent glacial-like features such as U-shaped hanging valleys and truncated spurs (e.g. Cole and Young, 1983). However, Quaternary glaciation did not extend down to the elevation of Unaweep Canyon, and glacial deposits are lacking (Soreghan et al., 2007). An alternative hypothesis posits that the canyon was carved by glaciation in the late Paleozoic, and later exhumed by the ancestral Gunnison River (Soreghan et al., 2007, 2008, 2014, 2015). A pre-Quaternary glacial origin remains controversial, in part because the Uncompahgre uplift was equatorial during the late Paleozoic. Previous geophysical and drilling surveys (Davogustto, 2006; Haffener, 2015) suggested that the valley might be over-deepened but were inconclusive regarding the exact depths and the valley geometry. A recent approach focused on acquisition of high-resolution reflection seismic data in fall 2017 (Patterson et al., 2018a, 2018b), and these data are also the basis for the present study.

## 3 Acquisition

The 2.4 km long reflection profile crosses the canyon in of its widest parts along a 4WD road, except for its first and last few hundred meters (Fig. 2). Geophone installation, acquisition, and demobilisation was done within 2.5 days. Recording stations were equipped with 385 Reftek 'Texans' data loggers / 4.5 Hz 1C geophones and with 120 Fairfield ZLand 3C 5 Hz nodes at

a 5 m interval. The ZLand nodes recorded continuously, while the Texans were only active during daytime due to memory constraints. The shot spacing is 10 m in the northern part and 5 m in the southern part, where maximal over-deepening was expected. Along the 4WD road, the truck-mounted and nitrogen-pressured A200 P&S source (Lawton et al., 2013) was utilized. This source provided ample energy to record strong basement reflections from 400 – 600 m depth (Patterson et al. 2018a,

2018b; Fig. 3). Manual hammering with 18 lbs sledge hammer provided seismic energy off-road. For both the truck-mounted source and the sledge hammer shots, five individual blasts were stacked at each shot location. All shot times were synchronized to GPS time. Due to time constraints, the northern- and southernmost parts of the profile were shot simultaneously. Shooting was done on Saturday and Sunday to avoid seismic noise from the a nearby active gravel pit. The state highway 141 intersects the profile in the southern part. Traffic on this road was moderate (one car / truck every 1 to 5 minutes). All shot and receiver

locations were surveyed with high-precision RTK GPS.

The geometry of acquisition design was optimized for reflection processing, resulting in dense receiver and shot spacing. The usage of nodal instruments was driven by logistical constraints, including partly steep and rough terrain, and a tight operational schedule. Receiver deployment and shooting was essentially completed in two working days without prior scouting, which would not have been possible with a conventional cable-based system and a fresh crew of mostly untrained student helpers.

An additional advantage of nodal acquisition is the possibility of recording at all offset ranges. Therefore, low-frequency geophones were chosen deliberately to ensure registration of first arrivals at long offsets.

## 4 Active source data and processing

Reflection processing and interpretation is currently ongoing and initial results are presented by Patterson et al. (2018a) and Patterson et al. (2018b). Here we focus on first arrival travel time tomography. In general, the first arrivals are of high S/N

(signal-to-noise) ratio, and they are visible up to 1.5 km offset (Fig. 3). The transition from low-velocity (1000 – 1500 m/s) sediments to high-velocity (> 4000 m/s) basement is indicated at most parts of the profile by a distinct kink in the first arrival travel time curve. This two-layer structure is not as clear towards the northern end of the profile, where the basement crops out but still exhibits low velocities at short offsets. This is indicative of pronounced erosion and weathering effects. In the area of expected over-deepening, refracted arrivals from the basement (Pb) are missing, while first arrivals through the sediments (Ps)

occur over longer offsets.

Overall, 18,263 sediment (Ps) and 16,104 basement travel time arrivals (Pb) are picked from the shot gathers. Signal processing is limited to bandpass filtering (10-30-130-160 Hz) and Automated Gain Control (AGC). Travel time picks have been validated by their reciprocal counterparts, wherever possible. Pb travel times represent refractions from the top of the consolidated basement, and Ps travel times represent both sediments and weathered basement. Both Pb and Ps picks are integrated into one

combined first arrival time pick set. In case of overlap (<0.1% of all picks), the minimum of Pb and Ps is designated as the first arrival.

3D first arrival travel time tomography is performed with the back-projection method of Hole (1992). Tests showed that a simple depth-dependent initial velocity model leads to poor data fit and partly unrealistic velocities (> 7000 m/s) in the southern part where the valley is expected to steepen. Therefore, we create a 2.5D initial velocity model from localized 1D inversions of CMP-sorted travel times. Using this improved initial model, the 3D travel time inversion converges to the final model shown

in Fig. 4 after 9 iterations. Offset restrictions and smoothing filters are successively relaxed to build a detailed yet robust model from top to bottom. The RMS travel time error of the final model is 0.03 s. The velocity model is indicative of over-deepening in the southern part, where high basement velocities are missing. This is in accordance with the lack of Pb observations in the shot gathers. Fig. 4 also includes a preliminary result of reflection imaging (depth-converted Kirchhoff Prestack time migration; Patterson et al. 2018b) which allows unambiguous interpretation for a U-shaped bedrock topography along profile

distances 1600 m – 2000 m. The U-shape is in alignment with the concept of over-deepening caused by glacial carving, which is also indicated by multiple bedrock-parallel reverberations. These are attributed to out-of-plane reflections from a bedrock dipping perpendicular to the profile direction, e.g. along the longitudinal axis of Unaweep Canyon (Patterson et al. 2018b). Significant longitudinal depth variations are further suggested from previous geophysical and drilling campaigns as well as from downstream basement outcrops (Davogustto, 2006; Soreghan et al., 2007; Haffener, 2015; Soreghan et al. 2015).

Interpretation of exact basement depths in smooth tomographic models is ambiguous due to inherent blurring of first-order velocity discontinuities. Therefore, the Pb travel times are also subjected to a delay time decomposition approach (Telford et al. 1990), providing the refractor structure in terms of delay times $td$ and refractor velocities $v_R$:

$$t(x) = \frac{x}{v_R} + td_s + td_G \tag{1}$$

In equation (1), $t(x)$ represents the picked Pb travel time at a specific offset $x$. $v_R$ is the refractor velocity and $td_S$ / $td_G$ are the source and geophone delay times, respectively. Observing multiple shots at the same geophone locations leads to an overdetermined linear equation system which is solved for $v_R$, $td_S$, and $td_G$. The delay time equation system can be generalized for laterally variable refractor geometry (Iwasaki, 2002). For a given vertical overburden velocity profile $v(z)$, refractor depths

$D$ and delay times $td$ at a specific location are related by equation (2):

$$td = \int_0^D \sqrt{\frac{1}{v(z)^2} - \frac{1}{v_R^2}}\, dz \tag{2}$$

In equation (2), $v(z)$ is taken from the first arrival tomography velocity field (Fig. 4) after capping velocities at 1800 m/s to account for the blurring towards basement velocities. The obtained refractor depth coincides on average with the 2900 m/s -

isoline in the first arrival tomographic model at most parts of the profile, as well as with the strongest gradient in this velocity field. The delay time solution is less reliable at the northern end of the profile where the assignment of Pb travel times is more challenging due to a more variable refractor velocity. This is possibly caused by significant shallowing and outcropping of the basement, which in turn leads to stronger weathering effects, resulting in a more gradual velocity increase with depth. At the southern end, Pb travel time assignment is also difficult due to the steep dip of the refractor. In the over-deepened section, the lack of Pb travel times and large refractor dips prohibit delay time inversion. Refractor velocities range between 4300 m/s and 5600 m/s, with the lowest values in the center of the northern flat section. Considering the laterally varying reliability and resolution of the three approaches (travel time tomography, delay time modelling, reflection imaging), we manually build a combined interpretation of the consolidated basement (white line in Fig. 4).

## 5 Passive data and processing

The Texan data loggers recorded continuously during day time, and the ZLand nodes also recorded during night. Thus, a significant amount of passive ambient noise data was acquired in addition to the active data. It is tempting to use interferometric techniques (Wapenaar, 2010a; Schuster, 2010) to recover surface waves traveling between receivers from the ambient noise field. Observed surface wave dispersion can be inverted for vertical variation of shear wave velocity structure. At local scales with dense receiver spacing, most commonly phase velocity dispersion is obtained from Multi-Channel-Analysis (MASW; Xia et al., 1999). Data recorded at larger and irregular receiver spacing is subjected to the Frequency-Time-Analysis (FTAN; Bensen et al., 2007; Levshin et al., 1989; Hannemann et al., 2014) which provides group velocity dispersion.

The acquisition was performed during the seismically quiet weekend days to obtain high S/N ratio for the active data. Ambient seismic noise interferometry requires noise sources in order to reconstruct the waves traveling between receiver stations. Traffic on state highway 141 is moderate, but nonetheless contributes to the ambient noise spectrum. Two large 4WD trucks were used for deployment and transporting the source, and their movements along the profile also generate surface wave energy. Many other studies find traffic noise to be a dominant ambient noise source at local scales (Behm et al., 2014; Riahi and Gerstoft, 2015; Chang et al., 2016), and specifically designed surveys are used for traffic noise imaging in urban areas (Cheng et al., 2016). For our data set, active shooting during the day is also regarded as a major contributor to the ambient seismic wave field.

A comparison of those different noise sources in the FK-domain is shown in Fig. 5. Ground roll can be discriminated from air waves by its dispersive characteristics. Figs. 5a,d show the effects of the acquisition truck moving at profile distance 1800 m and of an additional vehicle at HW 141 (starting at ca. 23 seconds). Both excite Rayleigh waves in the frequency range 2 – 15 Hz. Walking noise is initiated in the northern part of the profile. Non-dispersive sound waves from a passing thunderstorm are visible in Figs. 5b,e. Blasts from the truck-mounted source provide clear and dispersive surface waves (Figs. 5c,f), but lack energy at the low end of the spectrum (< 3 Hz). Since the penetration depth of surface waves is indirectly proportional to their frequency, the contribution of traffic noise (Fig. 5d) enables to increase the investigation depth of surface wave inversion. The

shot in Figs. 5c,f is located at the switch from ZLand recorders (N) to Texans (S). To some extent, this allows to compare the responses of the ZLand deployment with the Texan deployment, as the latter will dominate the positive velocity branch in the FK-transform. The apparently poorer response at higher frequencies is partly attributed to the local geologic situation, as the ZLand deployment coincides with the transition to outcropping and weathered basement. Additionally, tight coupling of the bulky and relatively top-heavy 3C - ZLand recorders to the ground is more difficult to achieve than for the conventional 1C geophones. Finally, the Texan deployment stacks more effectively in the FK-transform due to the larger number of instruments.

## 5.1 Interferometry

Processing of the continuous data aims at deriving a 2D shear wave velocity model from the dispersive Rayleigh surface waves which are obtained from interferometric processing. Since most of the stations were equipped with 1C geophones (Texans), we use the vertical component data only and extract Rayleigh waves. As both the active shots and the ambient traffic noise excite Rayleigh waves (Fig. 5), we do not separate these data domains but instead use all data from the entire recording period. The workflow starts with cutting the continuous data into 30 seconds long time windows. We don't require instrument simulation, as the natural frequencies of the two types of geophones used are very close (4.5 Hz and 5 Hz, respectively). Pre-processing is limited to temporal normalization (1-bit normalization; Bensen et al. 2007). Spectral whitening is not applied since it is an intrinsic part of the following cross-coherence method used for the calculation of the interferograms. Tests with substituting 1-bit normalization by Automated Gain Control (AGC) did not result in significant changes in the interferograms. Interferogram calculation follows the virtual source method (Bakulin and Calvert, 2006), e.g. each 30 seconds long time window of each receiver station is cross-correlated with the corresponding time window of all other stations. The cross-correlation $G_{AB}(f)$ between a receiver station B and a virtual source station A is calculated in the spectral domain by equ. (3):

$$G_{AB}(f) = \frac{X_B(f) \cdot \overline{X_A(f)}}{\|X_B(f)\| \cdot \|X_A(f)\| + \varepsilon^2} \tag{3}$$

Equ. (3) is a measure of cross-coherence (Aki 1957; Prieto, Lawrence and Beroza 2009; Wapenaar et al. 2010b). In equ. (3), $X_A(f)$ and $X_B(f)$ denote the Fourier transformation of the recorded and pre-processed data at stations A and B, respectively. The overbar denotes complex conjugation. $\varepsilon$ describes a stabilization term in case the product of the amplitude spectra approaches zero, and it is chosen as 1% of the average amplitude spectra. The interferogram in the time domain is obtained from the inverse Fourier transformation of $G_{AB}(f)$.

For each virtual source-receiver pair, the individual correlations of all 30 seconds long windows are stacked into one final interferogram. Finally, 486 virtual source gathers are obtained (Fig. 6). The gathers show clear move-outs with varying velocities in different frequency ranges, and with energy being distributed in the frequency range 2 – 15 Hz. The characteristics of the causal and acausal parts indicate that the main source of the ambient noise is located towards the south, and traffic from state highway 141 appears to be a significant contribution. Virtual source station 12040 (bottom panel in Fig. 6) is located

directly at the road, but still most of the stations southward exhibit dominant acausal surface waves, indicating noise sources being located even further to the south. Besides road traffic and movements along the acquisition line, no other natural or anthropogenic activity is expected to generate seismic noise in the observed frequency band in this widely unpopulated region. Additional noise might be presented by reflected surface waves related to the steeply dipping mountain front in the south. This front might backscatter seismic energy generated at the road and within the acquisition line towards the north. Observation of reflected low-frequency earthquake surface waves are reported by Stich and Morelli (2007), and scattered and reflected surface waves are common in exploration settings (Strobbia et al., 2011; Halliday et al., 2007, 2010). Behm et al. (2017) speculate on reflected high-frequency surface waves as ambient noise sources from data acquired in a local network on an East Greenland glacier. They also identify the steep basement cliffs as potential reflectors with providing and impedance contrast to the ice, and their environmental settings (e.g., limited anthropogenic and natural sources) are similar to this study. However, specific geometric relations between the noise -source(s), the reflecting surface, and the acquisition geometry are required to explain the absence of causal arrivals at the same time. A more detailed view at the causal arrivals at the southernmost stations shows offset-independent move-outs with very high to infinite apparent velocities at some stations. Such behaviour can be caused by non-stationary noise sources, and indeed a driveway oriented perpendicular to the profile orientation was used to access the southern end of the profile. We therefore suggest that driving along this off-profile road contributes to the ambient noise spectrum in this part of the profile.

## 5.2 Inversion for S-wave velocity structure

The observed dispersion of surface waves in the virtual source gathers is inverted for the 2D shear-wave velocity structure along the profile. We start with subdividing the profile into 25 100-m-long sections and perform source-receiver sorting of the interferograms accordingly. All interferograms which have their virtual source and receiver station within one section are assigned to this section. Within each section, all interferograms are stacked in 5 m – (absolute) offset bins, resulting in one virtual shot gather representative of that section. By this approach, we take advantage of the multi-fold coverage while still maintaining lateral resolution, and attenuate effects of the topography on the surface wave propagation (Köhler et al., 2012; Ning et al., 2018). Subsequently, each stacked virtual shot gather is subjected to surface wave phase velocity dispersion analysis, dispersion curve picking, and inversion for vertical shear wave velocity structure. This corresponds to the classical MASW workflow (Multichannel Analysis of Surface Waves; Xia et al., 1999).

We employ the wavefield transformation method of Park et al. (1998) to image dispersion of the spectra of the surface waves. We follow the energy peak to automatically pick the multimodal dispersion curves. Considering that the higher mode dispersion curves only exist in a few sections, we pick the fundamental mode dispersion curves only. We further resample the picked dispersion curve to ensure the efficiency of inversion as well as the coverage of multiple wavelengths. In this case, we resample the lower frequency (< 8 Hz) part dispersion data along the wavelength axis with a 50 m sampling step, and the higher frequency (> 8 Hz) part along the frequency axis with a 2 Hz sampling step.

The picked and resampled phase velocity dispersion curves are inverted for 1D shear wave velocity profiles $V_S(z)$ based on the classical damped least-square method and singular-value decomposition technique (Xia et al., 1999). We use P-wave velocities from the travel time tomography model (Fig. 4) and build the density model $\rho(z)$ from the P-wave velocities $V_P(z)$ with Gardner's relation (Gardner et al., 1974):

$$\rho(z) = 0.31 \cdot V_P(z)^{0.25} \tag{4}$$

Density is needed as one of the model parameters ($V_P$, $V_S$, $\rho$, thickness) for the inversion, since the Rayleigh wave velocity is a function of these parameters. Surface wave phase velocity has low sensitivity on density (Xia et al., 1999), therefore usually
just constant densities are chosen for the inversion. Recent research show that inappropriate use of constant density can lead to overestimation of the surface wave velocity and can introduce model artefacts such as a low-velocity layers (Ivanov et al., 2016). Gardner's relation, even though it might overestimate densities in unconsolidated structures, is already a significant improvement to commonly used constant densities. We set the maximum inversion depth to be half of the obtained maximum wavelength for each dispersion data. In general, this method is fast and stable, and most inversions could be completed within
6~7 iterations with a minimum root-mean-squared error at ~20 m/s. This error represents the misfit between the picked and predicted surface wave velocities.

Fig. 7 presents examples of stacked virtual shot gathers (left panel), the measured and picked dispersion spectra (middle panel), and the inverted $V_S(z)$ functions (right panel). The clear dispersion curves indicate a high S/N ratio of the stacked virtual shot gathers. The virtual shot gathers refer to three locations (profile distances 350 m, 1150 m, 1950 m) as shown in Figure 8. The
dispersion spectra shows energy being distributed from 2 Hz to more than 35 Hz. We can also detect the air wave energy in the dispersion spectra in Fig. 7b where the yellow line indicates a velocity of 340 m/s. The cyan curves indicate the final dispersion curves used for inversion, where the error bar represents the width of the amplitude spectra which is used as a weight in the inversion. The white dashed lines indicate the sampling power of the virtual shot gathers ranging from the maximum to the minimum wavelength:

$$\frac{1}{\lambda_{min}} = \frac{1}{2 \cdot dx} \ , \quad \frac{1}{\lambda_{max}} = \frac{1}{L} \tag{5}$$

In equ. (5), $dx$ and $L$ refer to the geophone spacing (5 m) and the maximum offset (100 m), respectively. Therefore, the maximum and minimum wavelengths calculate to 100 m and 10 m, and we set the upper limit of the frequency range for the
picked dispersion curves to be ~35 Hz. A conservative rule of thumb used in active surface wave survey suggests that the ratio $r$ of the minimum array length $L$ to the desired maximum wavelength $\lambda_{max}$ should be in-between 1.5 and 2.0 (Xia et al., 2006; Foti et al., 2018).However, Park and Carnevale (2010) show that the maximum error in phase velocity retrieval is less than 5% at wavelengths $\lambda$ for $L \leq \lambda \leq 2L$, which means an optimal $r$ could be between 0.5 and 1.0. Pasquet et al. (2015a,b) argue that

dispersion curves can be used down to low frequencies where the spectral amplitude is becoming too weak. In that direction, a range of other studies find that the ratio *r* can be significantly smaller than 1.5 while still providing meaningful results (e.g. O'Connell et al., 2011, r=0.32; Pasquet and Bodet, 2017, r=015; Zhang et al, 2019, r=0.25). The identification of maximum wavelengths varies also with the data quality, dispersion measurement techniques (Luo et al., 2008), source-receiver

configuration (Park and Shawver, 2009) and processing techniques (Zheng and Hu, 2017). In summary, these studies reflect the commonly accepted knowledge that array length is not the only factor which determines the maximum wavelengths to be recovered from MASW techniques.

We chose the minimum frequency as 3.5 Hz due to the high-quality data and clear dispersion curves which appear meaningful down to frequencies as low as 2 Hz. Depending on the velocity, this results in minimum wavelength-profile length factors

between 0.3 and 0.7. In Fig. 7 we observe that the dispersion signature at the location X=1950 m is different from the two other ones and indicates a velocity inversion with depth (Shen et al., 2017).

The 25 $V_S(z)$ functions are assigned to the centre of their corresponding 100 m long sections and are interpolated along the profile (Fig. 8). We observe the same large-scale structure as derived from the active source processing, e.g. thickening of the low-velocity surface zone towards the south, lack of high velocities and decreased penetration depth in the over-deepened part,

and high velocities close to the surface at the southern end of the profile. A significant discrepancy is the apparent increase in dip of the basement at the profile distance ~900 m when compared to the basement interpreted from active source data. However, there is an indication of a basement velocity decrease in the tomographic P-wave velocity (Fig.4) model as well as in the refactor velocity model, and basement reflections in the shot gathers suggest a sudden local change in dip at this location. A buried basement fault or significantly fractured basement may explain this feature, but this is subject to further investigation.

The shallow S-wave velocity structure in the over-deepened section (profile distance ~1600 – 2100 m) is indicative of an inversion zone (see also Figs. 7h,i) and is discussed in more detail in the next section.

## 6 Discussion

Our discussion section is organized in three parts. First, we provide an overview on the expected sedimentary stratigraphy based on a core from a distant well. Secondly, we calculate the ratio of P- to S-wave velocities and attempt an interpretation

in the context of this expected stratigraphy and other studies in similar geologic settings. Lastly, we critically assess some aspects of our workflow and their impacts on our interpretation.

### 6.1  Local sedimentary stratigraphy

In 2006, two closely spaced wells were drilled in Unaweep canyon ca. 5 km eastwards of the seismic down to depths of 320

m and 329 m, respectively (Soreghan et al., 2007; Fig. 1), where only the deeper one penetrated basement. In the retrieved core of the sedimentary section, three distinct units were delineated, on the basis primarily of sedimentary facies and provenance (Soreghan et al., 2007; Balco et al., 2013; Soreghan et al., 2015). The uppermost ~160 m comprises clast- and

matrix-supported conglomerate, with clasts ranging from granule to cobble/boulder size, of both Precambrian basement and Mesozoic sand/siltstone. Local sandy/clayey interbeds also occur, all poorly indurated. This fanglomerate unit also crops out at roadcuts further down the canyon, and is of Pleistocene age. This unit transitions, through a ~7 m interval of carbonate-rich paleosols, to an upwardly coarsening interval of well-sorted, poorly indurated sand yielding to underlying silt and well-

compacted clay that extends to ~315 m depth. This Pleistocene unit is interpreted as lacustrine, with a provenance that includes volcanic lithics tied to the ancestral Gunnison River, in addition to Mesozoic sedimentary lithics. It was deposited 1.4 million years ago when a landslide on the western side blocked the ancestral Gunnison River feeding the lake from the east (Balco et al., 2013). The basalmost ~5 m of the core comprises a moderately indurated diamictite consisting entirely of Precambrian basement clasts encased in a fine-grained matrix, and inferred to be of Paleozoic age (Soreghan et al., 2007).

Given that the fanglomerate comprises the modern surface, we infer that this unit also occurs in the western canyon, underlying the seismic profile. The landslide blockage that impounded the ancestral Gunnison River is inferred to have occurred in western Unaweep Canyon (Balco et al., 2013), hence the lacustrine section should also occur in this location at the same elevation. Given that this location is 5 km more distal to the river source than the Massey core, we infer that the lacustrine section here should be finer in general, and thus contain a higher proportion of compacted clay at depth. Beneath the lacustrine section

Soreghan et al. (2007) posit the existence of an interval correlative to the Permian Cutler Formation (Werner, 1974; Soreghan et al., 2009) which is exposed at the western mouth of the canyon, and comprises a mixture of conglomerate, granule conglomerate, and silt/mudstone. The exposed Cutler Formation strata are very poorly consolidated and poorly sorted, and show signs of significant fluid alteration (Hullaster et al., 2019).

**6.2 Calculation and interpretation of the $V_P/V_S$ – ratio**

In geological settings, low seismic velocities are usually associated with poorly consolidated soils and rocks. This applies to both P- and S-wave velocities, although S-wave velocities are more affected due to their sole dependence on the shear modulus. The additional knowledge of the ratio of P- to S-wave velocities can help to further constrain subsurface properties. A sudden increase of the P-to-S velocity ratio with depth is often used as an indicator for the groundwater table (GWT) as shear-wave

velocities experience no significant change when pore space voids are filled with fluid. For near-surface soils (< 50 m depth), several studies can be found which report $V_P/V_S$ – ratios based on seismic surveys. This is largely because of the interest in shallow soil structure for geotechnical and hydrological applications and the ease at which shallow P- and S-wave data can be acquired. Uyanik (2011) summarizes $V_P/V_S$ – ratios of seismic measurements in shallow (< 20 m depth) saturated sediments (gravel, sand, clay-silt) with porosities ranging from 20% to 50%. For 100% water saturation, his data show $V_P/V_S$ – ratios

ranging from 3.3, to 7.2. Pasquet et al. (2015a) combine P-wave refraction, S-wave refraction, and surface wave inversion to image a shallow GWT (< 20 m depth) in a weathered granitic basement. They state low $V_P/V_S$ – ratios (<2.75) for the low-porosity/low-permeability granitic basement and higher ratios (3.0 – 4.0) for wet soil close to the surface.

In-between the shallow surface and deep crustal / reservoir targets, only a small number of studies report $V_P/V_S$ – ratios for intermediate depths comparable to our study. Konstantaki et al. (2013) derive hydrological and soil mechanical parameters

across the Alpine Fault in New Zealand. They apply P-wave tomography and MASW to data from active shot gathers and derive velocity models down to depths of 60 m. They find $V_P/V_S$ – ratios larger than 3.0 and up to 9.0 for wet sand, gravel, and silt lithologies, and were able to interpret the GWT from their results. Bailey et al. (2013) conducted a deep P- and S-wave reflection survey in a geologic setting comparable to our study. Their site comprises a several hundred meters thick sedimentary sequence of Quaternary sands and clays of Pleistocene age, which also includes lacustrine sediments. They were able to derive $V_P/V_S$ – ratios with high lateral and vertical resolution from the correlation of P- and S-wave reflections and from MASW. In the shallow surface (< 50 m depth), they find $V_P/V_S$ – ratios as high as 10, which were interpreted as soil pockets with high potential for liquefaction. The deep structure (50 – 500 m depth) exhibits $V_P/V_S$ – ratios between 3.0 and 6.0. Zuleta and Lawton (2012) present a similar dataset comprising multicomponent data with P- and S-reflections. They investigate a late Paleozoic sedimentary basin in British Columbia and derive $V_P/V_S$ – ratios between 6.0 at the surface and 2.0 in depths of ca. 300 m. Their velocities are comparable to our studies, e.g. $V_P$ is ranging from 1950 m/s to 2800 m/s, and $V_S$ is varying between 350 m/s and 1400 m/s.

We calculate the ratio of the tomographic P-wave velocity and the S-wave velocity models (Figs. 4, 8, 9). In order to account for the different parameterization of the travel time tomography and the dispersion inversion, we average P-wave velocities within each surface wave inversion depth layer before we take the ratio. In the left part of the profile, we encounter $V_P/V_S$ – ratios between 1.8 and 2.5 for both the overburden and the basement. Between profile distances 700 m and 1100 m, the migration shows a pronounced reflector in the depth range 50 m to 100 m which could potentially represent a GWT. There is however no significant correlation of the $V_P/V_S$ – ratio with this reflector. In case of the basement in the left part of the profile, $V_P/V_S$ – ratios larger than 2 and moderate P-wave velocities (4.0 – 5.5 km/s) are indicative of significant weathering and/or fracturing of the Precambrian granites. The $V_P/V_S$ – ratio changes to significantly higher values (3.0 – 6.0) in the over-deepened part of the profile. The top of this zone of high $V_P/V_S$ – ratios reaches the surface at the southern part of the profile, where West Creek occupies the lowest topographic point. The zone dips towards the north and its top is found at ca. 120 meters depth at the presumed northern edge of the over-deepened section. A northward dipping reflector is found in a comparable depth range in the seismic image, and also the P-wave velocities (1500 m/s – 1800 m/s) correspond to typical velocities of saturated near-surface sands and gravels (Knights and Endres, 2005; Everett, 2013). We therefore interpret the increased $V_P/V_S$ – ratio in the over-deepened section to represent water-saturated sediments. As with smooth P-wave velocities, it is not clear which value of the $V_P/V_S$ – ratio represents the exact threshold to delineate the GWT. The GWT itself might be imaged more accurately by the reflector, while the $V_P/V_S$ – ratio is used to estimate the thickness and lateral extent of the saturated zone. Since the dip of the interpreted saturated zone opposes the slope of the topography, this aquifer needs to be confined or it is leaking through fractured basement in the north. The latter hypothesis would be supported by the relatively low P- and S-wave velocities between profile distances 900 m to 1400 m (Figs. 4, 8).

Both the tomographic P-wave velocity model and the S-wave velocity model from the stacked gathers with a maximum offset range of 100 meters have only little penetration depth in the over-deepened section. To increase the investigation depth, we

extend the tomographic velocity model with interval velocities obtained from reflection processing (Patterson et. al., 2018b). The two velocity models are tied together at an elevation of 1800 meters, where a smoothing filter is applied to account for their different nature (smooth travel time tomography vs. discontinuous interval velocities). A deeper reaching S-wave velocity model is derived from stacking all source-receiver sorted interferograms between the profile distances 1500 m and

2100 m. The resulting maximum offset of 600 m allows for picking a dispersion curve with minimum frequencies around 1 Hz, which in turn results in a significantly larger penetration depth of the inverted S-wave velocity model (Fig. 10). For both P- and S-wave velocity models, the increase in investigation depth comes at the expense of reduced lateral resolution. However, at this stage we are primarily interested in a representative 1D section of the over-deepened part. To calculate $V_P/V_S$, we again average the P-wave velocities in the corresponding layer depths of the S-wave velocity model.

Fig. 11 shows a compilation of the 1D-velocity models in the over-deepened section. In general, the P-wave velocities in the range 1200 – 2700 m/s correspond to those established for other Pleistocene alpine valley fills (Brueckl et al., 2010; de Franco et al., 2009). In Fig. 11, we also show the sonic log from the Massey well. The well is located upstream West Creek and 5 km to the east of the seismic profile (Fig. 1), where the topographic elevation is also 80 m higher. The sonic log indicates a P-wave velocity decrease at an elevation of ca. 1830 m, which correlates with the transition from the fanglomerates to the

lacustrine sands. The merged seismic P-wave velocity profile shows a discontinuity at this elevation, which however also indicates lower velocities above the sand. This discrepancy can be explained by different local composition and compaction of the fanglomerate at the two locations. Another possibility for the difference is a variable groundwater table, leading to saturated fanglomerates at the well location and dry fanglomerate at the seismic profile. This is in fact supported by the $V_P/V_S$ – ratio, which is low (2.0 – 2.5) above the top lacustrine horizon and raises to significantly larger values (3.4 – 4.0) below. The

increase in P-wave velocities correlates with a decrease of S-wave velocities, which also suggests a vertical change of lithology. Overall, we interpret the high $V_P/V_S$ – ratios as an indicator for saturation in the lacustrine sands below the fanglomerate.

The last few meters of the core transit into a mixture of basement clasts and Palaeozoic sediments. This transition correlates with a velocity discontinuity in the interval P-wave velocities and the onset of a gradual increase of the S-wave velocities. The high P-wave velocities would suggest sediments other than clay or sands, which usually are characterized by velocities not

larger than 2200 m/s (Knight and Endres, 2005). Soreghan et al. (2007, 2008, 2014, 2015) speculate that the over-deepening of Unaweep Canyon was caused by glaciation in a late Palaeozoic icehouse, and that the lacustrine sands lie on top of an upper Paleozoic sedimentary fill which could explain higher seismic velocities.

The interval velocities were obtained from conventional velocity analysis and the Dix equation. Steep dips as the valley flanks can lead to an overestimation of the velocities in the deeper sections of the sediment fill. However, the extracted interval

velocities are located at the centre of the U-shaped valley cross section, where both reflections from the flanks and from the flat bottom do occur. Out-of-plane reflections are also present and can introduce non-physical layering in the velocity profile. Given these uncertainties, we do not attempt to correct individual stacking and interval velocities for dip and subsequently do not show or interpret $V_P/V_S$ – ratio below the top of the presumed Palaeozoic sediments at the elevation of 1600 m. Our main new insight from both P- and S-wave velocity models at larger depths is the identification of the top and bottom of the lacustrine

section, and a general increase of velocities below this section. Forward modelling of basement reflections could help to constrain deep interval velocities and subsequently $V_P/V_S$ – ratio, but this is beyond the scope of this study.

### 6.3 Methodological aspects

Our interpretation of the $V_P/V_S$ – ratio is based on velocity models of different origins and of different parameterization. Both the tomographic P-wave model and the S-wave model from dispersion inversion do not explicitly comprise distinct velocity discontinuities such as the prominent sediment-to-basement transition. This interface will be represented as a strong gradient in an overall smooth velocity field, and the corresponding $V_P/V_S$ – ratio will not allow for the exact definition of a groundwater table. Nonetheless, the lateral variation of the $V_P/V_S$ – ratio in the over-deepened section correlates with the seismic image and

the P-wave velocity model, and suggests the existence of an aquifer (Fig. 9). The $V_P/V_S$ – ratio does not give any indication for the transition from sediments to the basement in the northern part of the profile, even though both P- and S-wave models sample the basement at sufficient depth ranges. This can be indicative of significant weathering of the top of the Precambrian granite. However, we are also aware that subjective choices of parameters used in the surface wave processing and inversion sequence (minimum wavelength, dispersion measurement algorithm, or density, layer thickness, and P-wave velocity

constraints) will impact the final S-wave velocity model. Therefore we prefer to interpret significant contrasts in the $V_P/V_S$ – ratio only, e.g. such as the high values in the lacustrine sands.

The structural interpretation of the asymmetric valley structure and the steep and sudden dip at its southern rim is supported by both the P-wave and S-wave velocity models (Figs. 4, 8). Dispersion analysis also gives better evidence of velocity inversion zones than classical travel time tomography which is less sensitive to velocity decrease with depth. In our interpretation the

vertical trends of S- and P-wave velocities are partly decoupled due to water saturation.

The dense receiver spacing allows for relatively high lateral resolution of the S-wave velocity model through sorting and stacking in source / receiver and offset bins, which comes at the expense of a loss in investigation depth. Nonetheless, even with these short offsets the investigation depth is comparable to the P-wave travel time tomography using long offsets. This compares to the results of Pasquet et al. (2015a) who find larger penetration depths of surface wave inversion over S-wave

refraction. Improved S-wave velocity imaging and higher lateral resolution might be obtained from simultaneous inversion of adjacent source / receiver cells (Konstantaki et al., 2013), or by calculating group velocity dispersion between individual receiver pairs (Bensen et al., 2007; Hannemann et al, 2014). The latter approach would be applicable to irregular receiver spacing but requires automatization of dispersion picking in case of a large number of receivers.

Sorting and stacking using larger offsets enables imaging of significantly larger depths, if low-frequency seismic energy is

present. In our case, the inclusion of traffic-induced ambient seismic noise provides frequencies as low as 1 Hz, which extends the frequency spectrum of the active source (Fig. 5). Seismic interferometry and the virtual source method provide a very efficient approach to merge the contributions from different active and passive seismic sources without the need for data selection or tailored processing schemes.

# 7 Conclusions

We have combined active and passive processing schemes to derive P- and S-wave velocity models of an over-deepened alpine valley. Both approaches complement each other in several aspects: (1) The P-wave velocity model is used to constrain the shear wave velocity inversion; (2) Ambient noise sources extend the spectrum to lower frequencies, thus enabling the imaging of deeper structures; (3) Independently derived P- and S-wave velocity models allow to calculate the $V_P/V_S$ – ratio which adds significantly to the geologic and hydrologic interpretation.

The calculation and interpretation of the $V_P/V_S$ – ratio is challenged by different parameterization of the models, and subsequently by the different sensitivity to lateral and vertical variation of the seismic structure. Information on subsurface lithology is essential to derive robust conclusions on hydrological and geological properties, and wherever this information missing the interpretation remains ambiguous. In particular the calculation of S-wave velocities from surface wave measurements is still impacted by poorly quantified uncertainties, and future research is needed to address this topic.

Our dataset shows that a deployment period as short as 30 hours in an area with little anthropogenic and natural seismic activity still contains ample ambient noise. Much of this noise stems from acquisition down-time when the active source truck is moving. Scattering and reflection of surface waves generate secondary sources which contribute to stationary phase sources required for the application of ambient noise interferometry. Interferometry and the virtual source method naturally blend active and ambient seismic sources without a need for separation of the two data domains, which broadens the frequency spectrum and the investigation depth

Large-scale 3D seismic acquisition projects, as routinely performed in the energy sector or other industrial applications, involve tens of thousands of active receivers, and those experiments might take weeks to months to be accomplished. If nodes are used, then the sheer amount of passive data acquired with dense spatial sampling invite the application of processing workflows like our study. Given the simplicity and high degree of automatization, detailed and robust subsurface models can be obtained quickly and at marginal additional costs.

To our knowledge, our study is the first one to report on the variation of $V_P/V_S$ – ratios in sedimentary infills of alpine valleys. Combined with reflection imaging and geologic extrapolation from a distant well, the data suggest that Unaweep canyon hosts a significant aquifer as indicated by $V_P/V_S$ – ratios significantly larger than 3 over a vertical extent of at least 100 m. Since resolution and accuracy of the seismic data decrease with depth, we note that a recently funded drilling campaign will provide ground truth and allow for verification or falsification of our interpretations in the very near future.

Given the fact that quaternary sedimentary strata cover a large range of the continental US (Soller and Garrity, 2018), our results invite the application of $V_P$ and $V_S$ measurements in non-alpine regions as well. Many areas in the US mid-west are prone to droughts while at the same time facing increased urbanization pressure, and influences by climate change. Mitigating these effects requires substantially expanding our knowledge on the distribution and characterization of potential groundwater resources (Taylor et al. 2012).

**Author contribution**

M.Behm did the active and passive processing of the data and wrote the manuscript, except the parts noted below. F.Cheng provided processing and description of the steps in section 5.2, and also contributed to processing of the active refraction data data. A.Patterson processed the active reflection data and contributed to processing of the active refraction data. G.Soreghan provided section 2 in the manuscript, and contributed to the interpretation

**Competing interests**

The authors declare that they have no conflict of interest.

**Acknowledgements**

Texan data loggers were provided by the IRIS PASSCAL instrument center. Acquisition was done in cooperation with the Seismic Source Facility (SSF) based at University of Texas at El Paso. The dedication and competence of Galen Kaip, Steve Harder, and Jefferson Chang contributed greatly to a successful acquisition. The Gateway community is thanked for local support. The project is partially funded from NSF-EAR-1338331. Andy Elwood Madden, Kato Dee, and Jim Blattman are thanked for discussion and helpful advice. Florian Bleibinhaus and Erika Angerer provided helpful review comments.

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

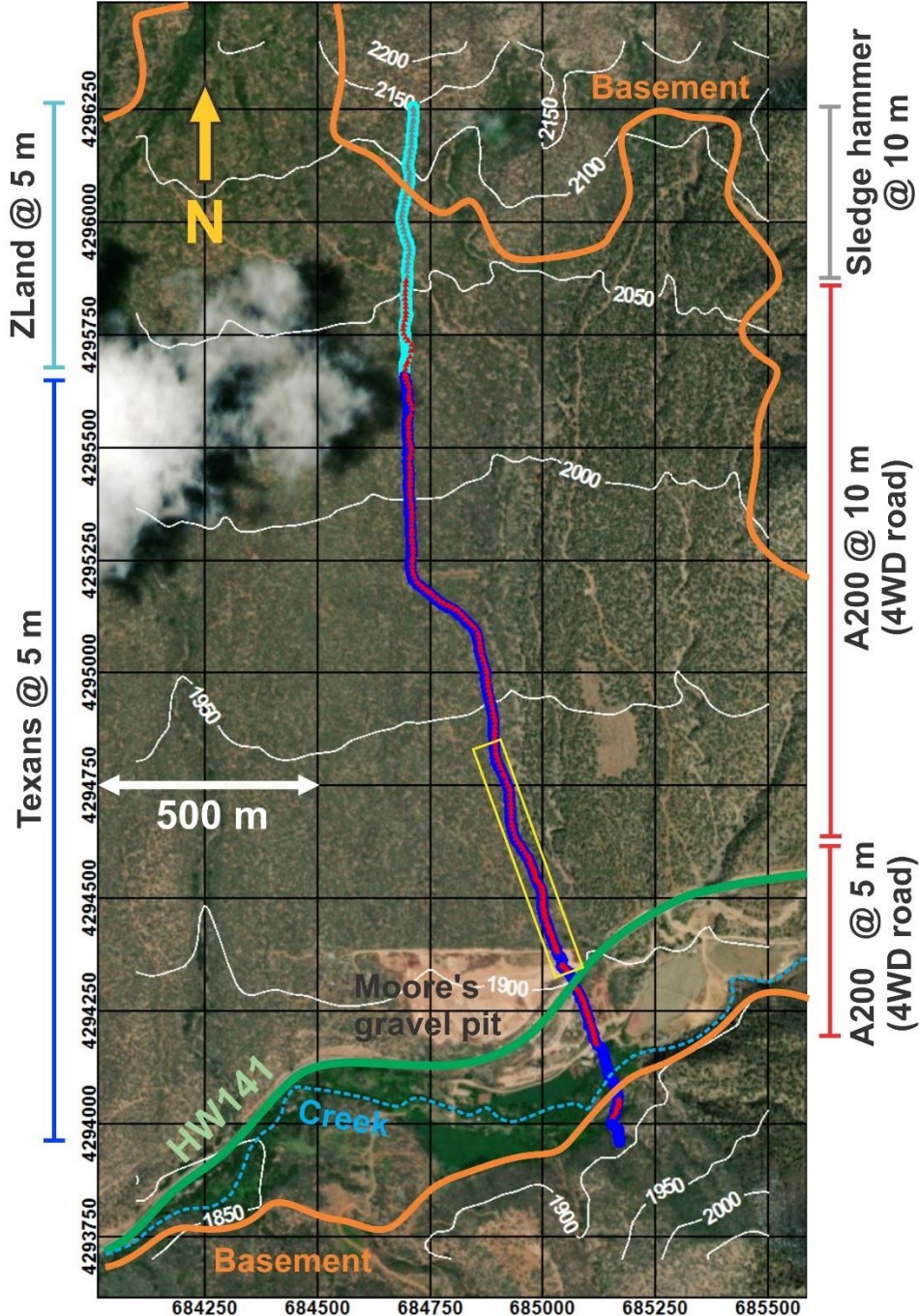

**Figure 2: Geometry of the 2017 seismic acquisition. Thick blue line: Texan 1C receivers; Thick cyan line: Fairfield 3C nodes; Red triangles: Shot locations of the A200 P&S source; Grey triangles: Shot locations of the sledge hammer; Green line: Road 141. White lines: Elevation contours in meters ASL. Blue dashed line: West Unaweep creek. The yellow rectangle outlines the representative area for Fig. 10.**

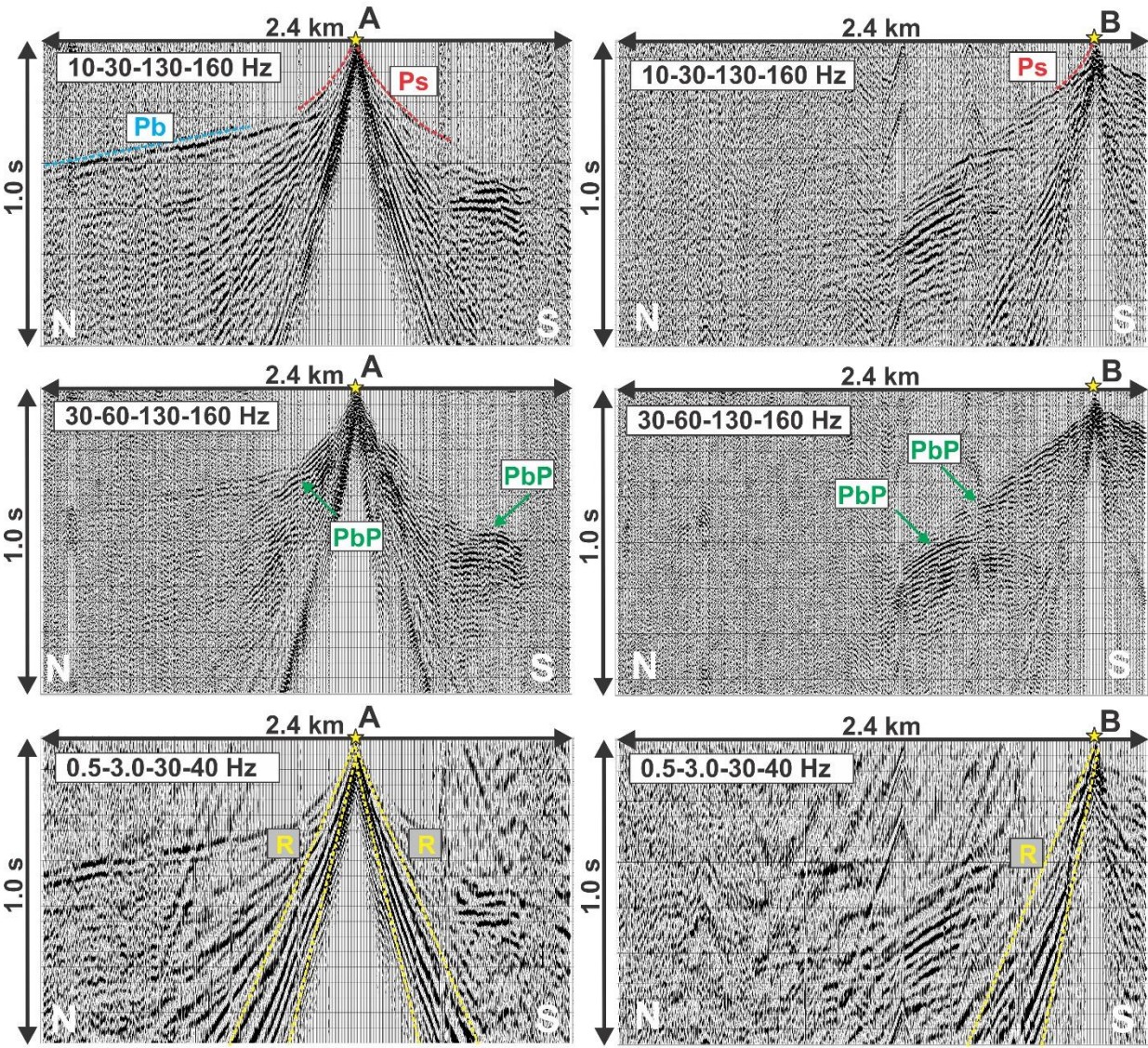

**Figure 3: Seismic data examples: Shot gathers A, B (location see Fig. 4) filtered in different frequency bands. Pb: refractions from the basement; Ps: refractions from the overburden (sediments); PbP: basement reflections; R; Rayleigh waves from the active source, but not also traffic-induced ground roll.**

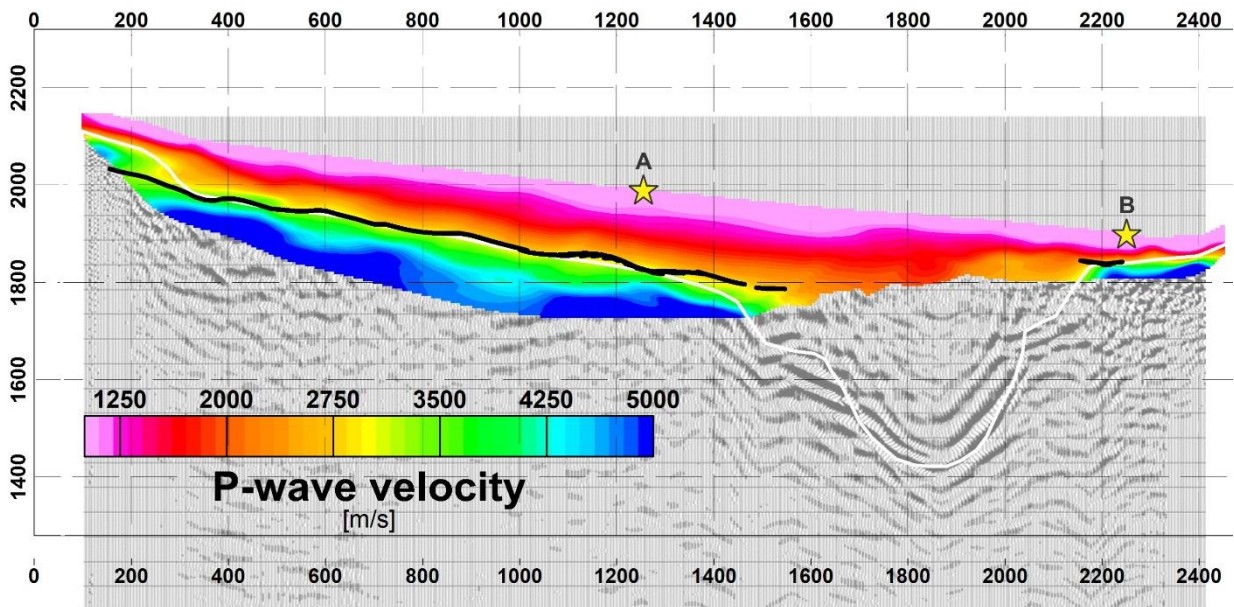

**Figure 4: P-wave velocity model obtained from travel time tomography. Backdrop is depth-converted prestack-migration (Patterson et al. 2018b). Black line: depth-converted delay time refractor. White line: Interpreted top of the consolidated Precambrian basement based on refraction and reflection data. A, B; location of shot gathers shown in Fig. 3.**

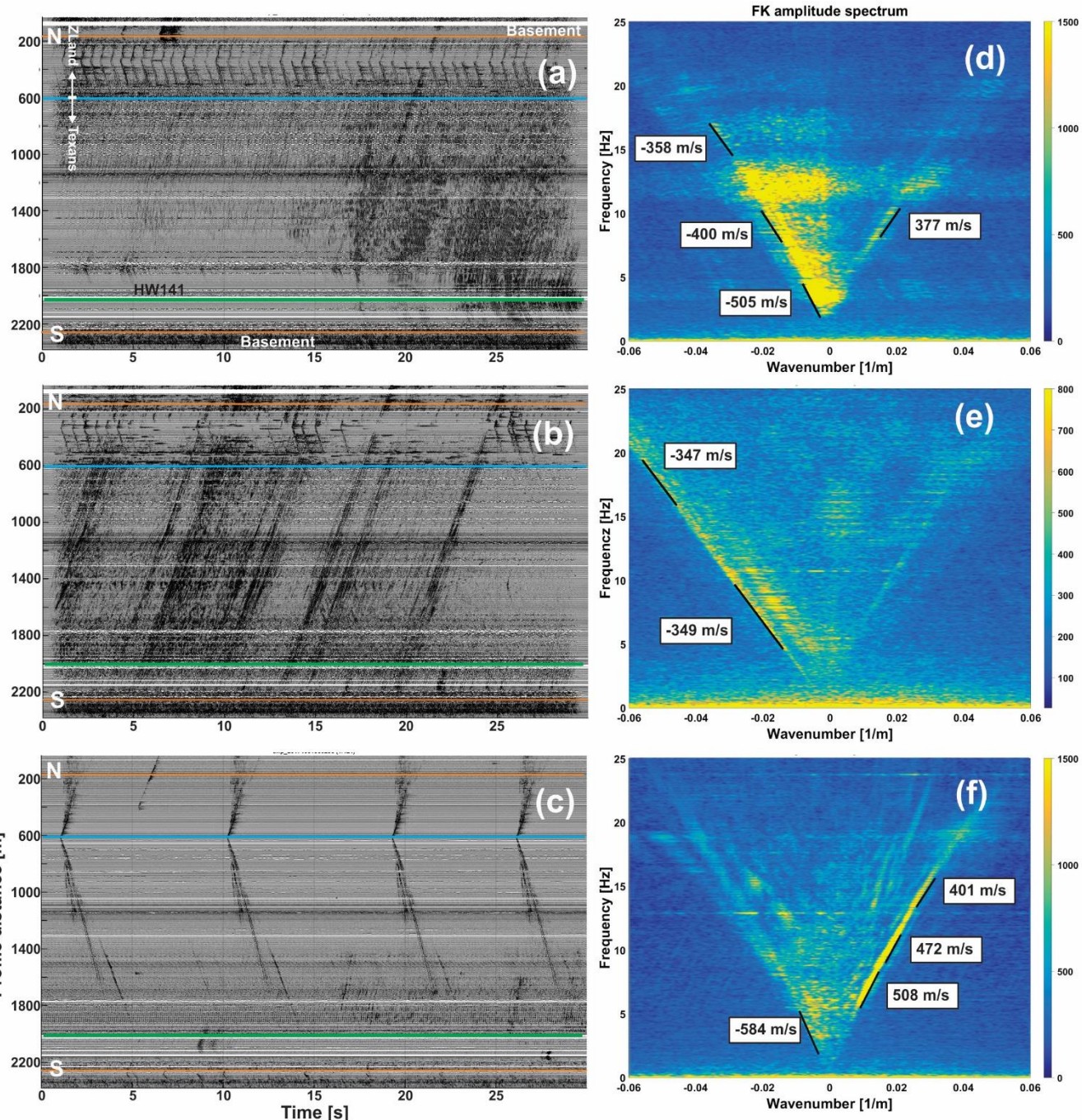

**Figure 5: Three 30-seconds long slices of continuous data (a-c) and their representation in the FK-domain (d-f). Traces are arranged horizontally from north (top) to south (bottom). Vertical axis: Profile distance. Blue line discriminates ZLand 3C nodes (north) from Texan 1C geophones (south). Green line: Road 141. Measured slopes in the FK gathers represent group velocities. (a,d): Traffic and walking noise. (b,e): Passing thunderstorm. (c,f): Succession of several blasts from the A200 source.**

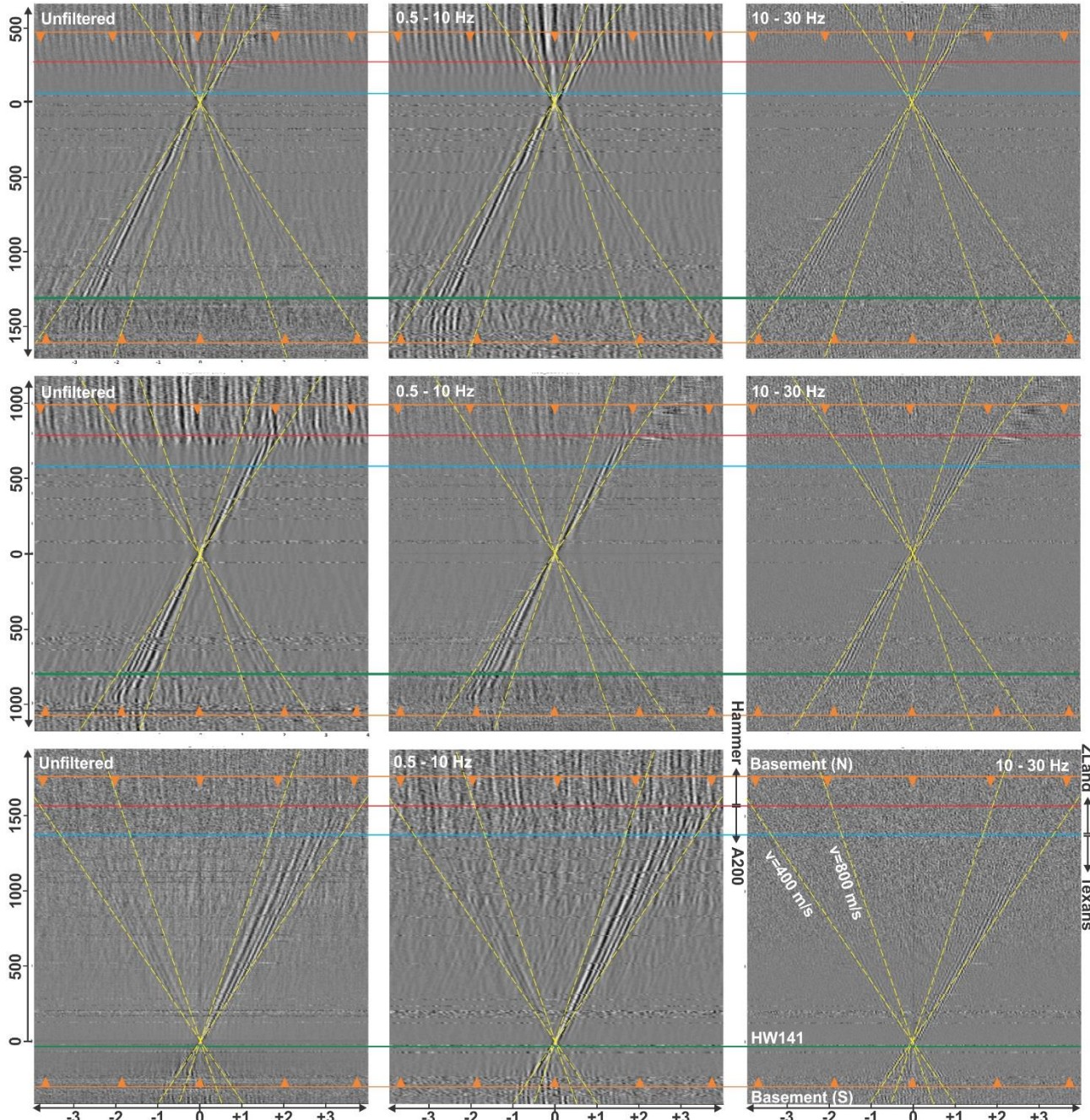

**Figure 6: Three virtual source gathers in three different frequency bands. Traces are arranged horizontally from north (top) to south (bottom). Vertical axis: Virtual source – receiver offset. Blue line discriminates ZLand 3C nodes (north) from Texan 1C geophones (south). Red line discriminates area with hammer shots (north) from area with A200 blasts (south). Green line: Road 141. Yellow lines: Moveouts for velocities of 400 m/s and 800 m/s, respectively.**

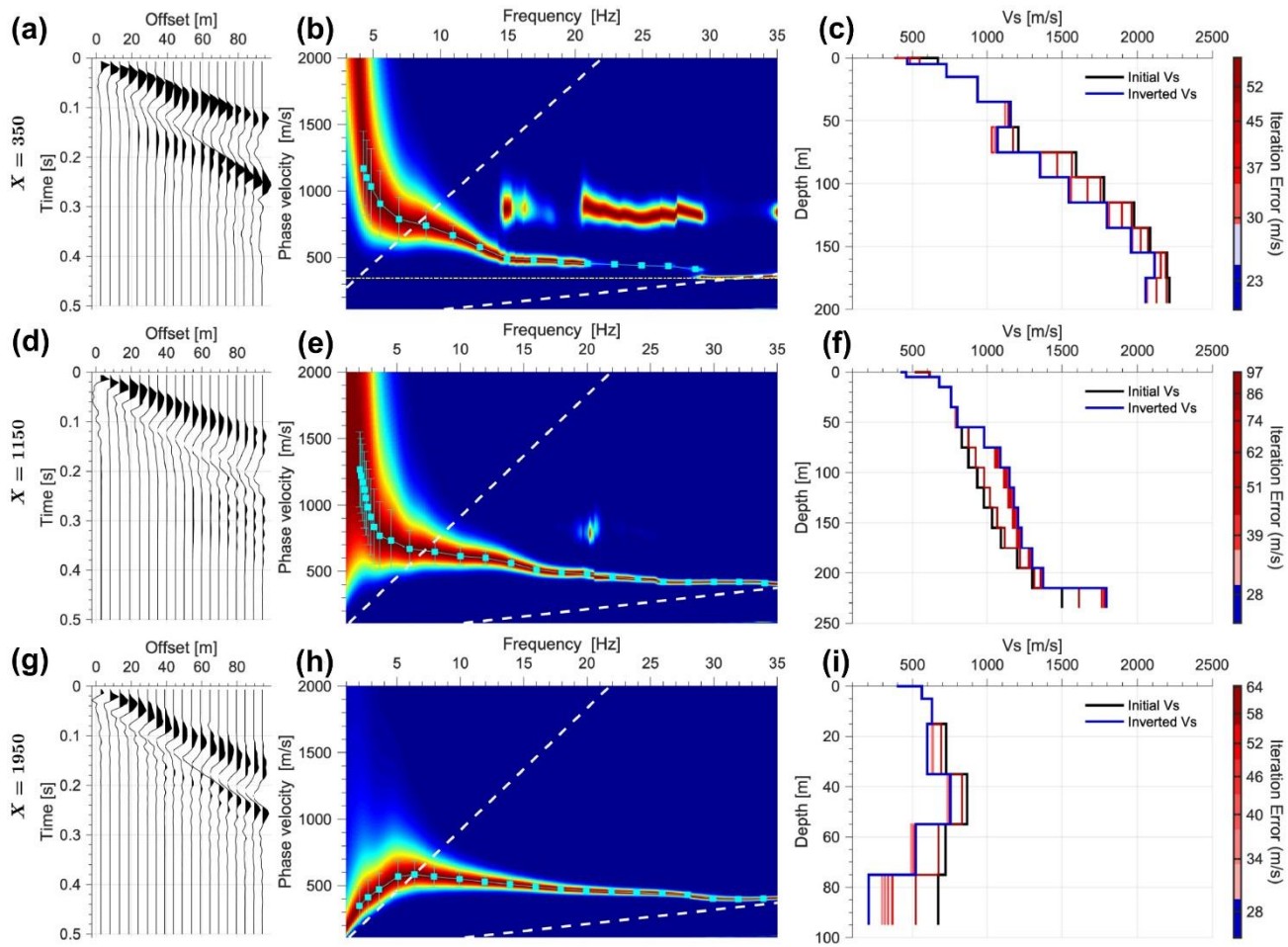

**Figure 7: Three examples of dispersion curves obtained from source / receiver sorting and offset stacking. Each dispersion curve is representative of a 100 m long section along the profile (see Fig.8 for location). Left panel (a,d,g): Stacked virtual source gather; Middle panel (b,e,h): dispersion curves and picks. (c,f,i): Inverted Vs(z) for all iteration steps. Blue curve (lowest data misfit) represents the accepted final model.**

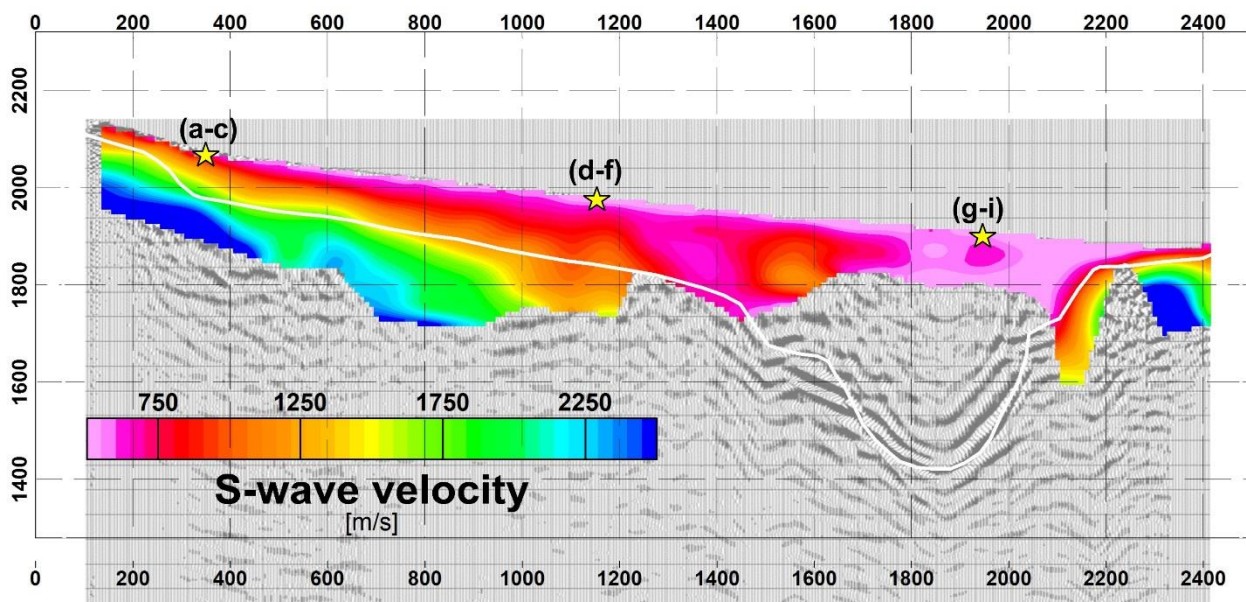

**Figure 8: S-wave velocity model obtained from interpolation of local 1D shear wave velocity profiles. Stars: Location of the corresponding 1D-inversions shown in Fig. 7. White line: Interpreted top of the consolidated Precambrian basement based on P-wave refraction and reflection data.**

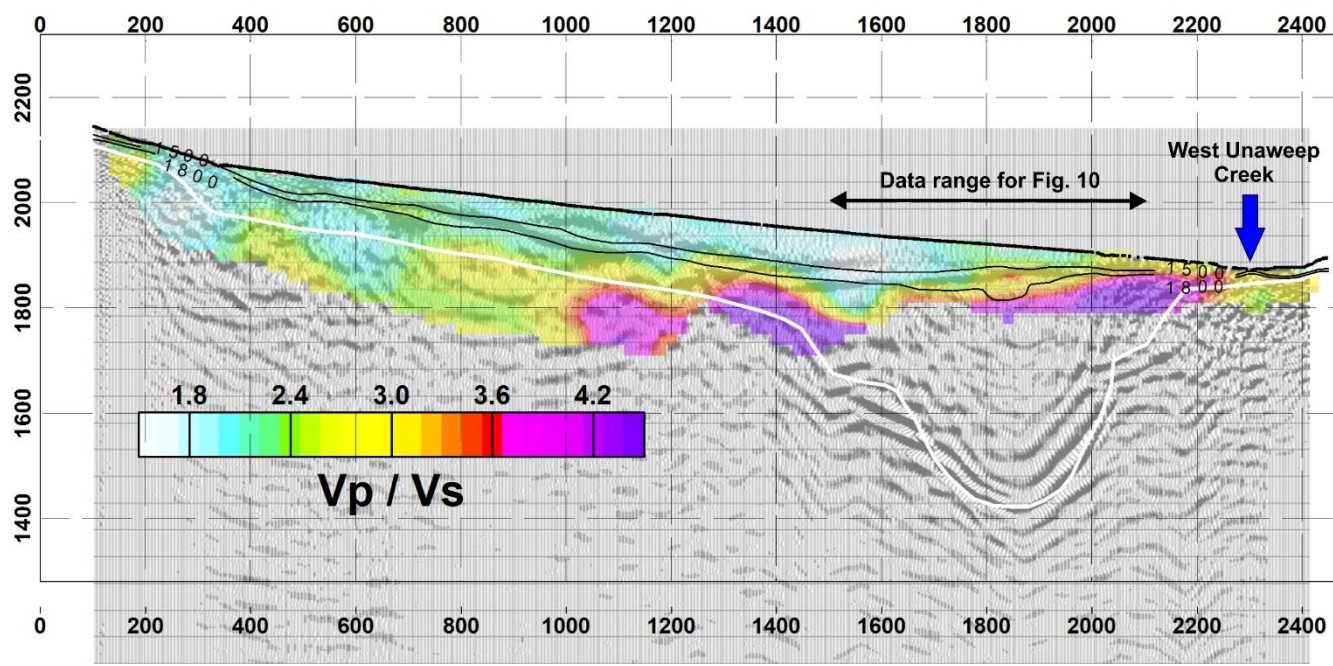

**Figure 9: Vp/Vs ratio. White line: Interpreted top of the consolidated Precambrian basement based on P-wave refraction and reflection data. Thin black lines: Contour lines of the P-wave velocity model (Fig. 4) for 1500 m/s and 1800 m/s.**

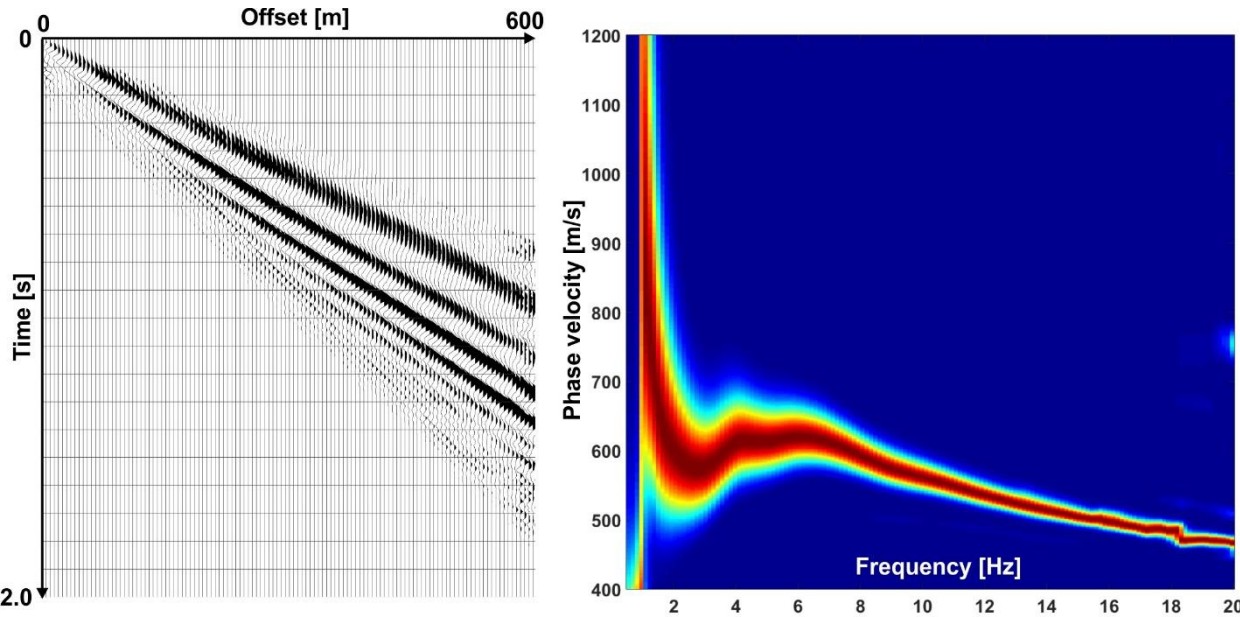

**Figure 10: Dispersion curve obtained from source / receiver sorting and offset stacking of all virtual source gathers within the over-deepened section (profile distance 1500 – 2100 m).**

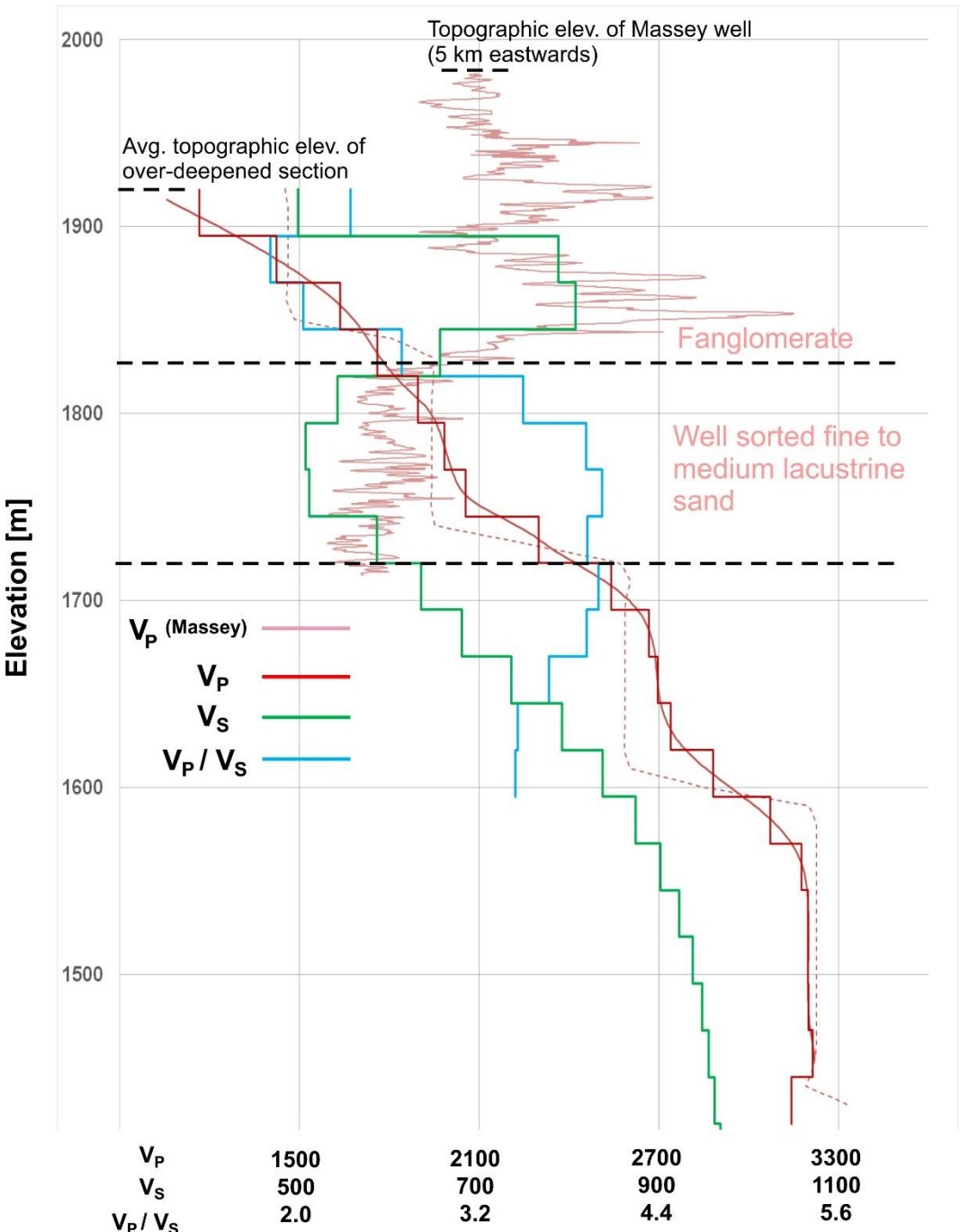

**Figure 11: Compilation of several 1D velocity models representative of the over-deepened section. Dashed red line: Interval P-wave velocity model from reflection processing (Patterson et al., 2018b). Smooth solid red line: P-wave velocity model from combination of interval and tomographic velocities. Solid red stair case line: Averaged P-wave velocities used for $V_P/V_S$ calculation. Green stair case line: Shear wave velocities from dispersion inversion. Blue stair case line: $V_P/V_S$ – ratio of the upper section. Lithologic interpretation and sonic log (bright red line) are from the Massey well located at 5 km distance to the seismic section.**