# Peer review of "Passive processing of active nodal seismic data: Estimation of $V_P/V_S$ - ratios to characterize structure and hydrology of an alpine valley infill"

_Solid Earth, 2019_

## Referee Comment (RC1) · Florian Bleibinhaus (Referee) · 2 Apr 2019

Comments on Behm et al., "Passive Processing… alpine valley infill"

It seems to me that the topic is well suited for the journal. The study presents new data on the seismic structure of a 200-500-m-deep Alpine valley, and it uses a novel combination of active and passive data analysis. The paper is well structured and, for the most part, clearly written and in good English. However, there are a few unclear points, that require a revision of the paper.

(1) The authors note a big difference between the two sensor types they use: 4.5-Hz 1-C-geophones of unknown making that "appear to have a better response than the ZLand 5-Hz 3-C-stations". It's unclear what exactly "better response" means (the frequency response functions are not shown) but the ZLand data in Figure 5 simply shows no useful signal. I am quite surprised by that, and I know colleagues who equipped themselves with ZLand nodes because of their supposedly great sensitivity especially at low frequencies. To claim that those instruments perform poorly has a potentially large impact on the manufacturer, and such a claim must be well founded: The authors must do more to reveal the source of the apparently poor data quality of the ZLands! Is it possible that no instrument simulation was performed? The authors don't write about it. It might explain the apparent differences. (It would also mean that the cross-correlations and the subsequent MASW would have to be re-done with the corrected data.) Or could it be a problem with the time-base?

(2) I find the interpretation section difficult to follow both how it is written and the conclusions it reaches. This paragraph should be rewritten for more clarity. I suggest the authors start this section summarizing all geologic information they have on the lithologies to be expected in this valley, including the geologic map and the well.
More importantly, I find the lithologic description (clay/mud below ~1720 m in Figure 11) to be inconsistent with the reported P-wave velocity of 2700-3300 m/s. In their book chapter "Rock physics principles for Near Surface Geophysics" (In: SEG-Investigations in Geophysics No. 13, 2005), Knight and Enders report P-wave velocities for clay to be at most 2200 m/s. Indeed, all technical literature I am aware of specifies P-wave velocities for loose materials significantly below 2700-3200 m/s. If real, such velocity indicates lithified rocks, which might be an important finding. If not real, and if the material is indeed clay, it contradicts the statement that it is part of a 400-m-thick aquifer (13/25).
Also, I find the interpretation of the GWT difficult. I fully understand that there are many reasons why $V_P$ might not be indicative for the GWT but not in this case of unconsolidated sand, where one would expect a sudden increase of $V_P$ at the GWT from maybe 800-1200 to ~1700-1800 (as seen in the well). It would be appropriate to make a first interpretation of the GWT from a contour line in the range ~1500-1800 m/s, and then check, if such a contour line coincides with the interpretation of the GWT from $V_P/V_S$-ratios.

(3) Please add some more detail about how the reflection-processing-based Vp-velocities were derived. (At an angle of 45°, as can be seen in the reflection section, NMO velocities would be 30% increased just from the dip, not to mention the potential distortions from the Dix formula.)

Minor remarks:

2/32 – In their model, resistivity is increased for the aquifer.

3/28 – I cannot see that road on the map.

5/9 – Over-deepening is an effect along the river-bed. How can you identify it from a cross-section?

7/28 – 8/2 Your explanation for the observation would still require that the reflections were stronger than the incident waves.

8/10 – "refrain" – you mean this cancels out through stacking?

8/23 – Please clarify what you need density for. Maybe it's not so important but for unconsolidated saturated sediments, Gardner's relation tends to significantly overestimate the density.

9/13 – I agree that the ratio profile length to wavelength should be at least 1.5-2. But I fail to see how you can then say that it's supposedly okay to use a ratio of less than 0.5. How does the overall length of your profile change the length of your subprofiles to which you apply MASW?

6/25ff (Interpretation) – Overall, I don't understand what the authors want to say in this paragraph: In the beginning the argue that there is a systematic trend regarding Vp/Vs and pore fill, and then they discuss examples that all appear to contradict those trends. Also, references to lower crustal studies, or studies where the GWT is in fractured granite, should be avoided. It's not enough for a general overview, and too much for loose sand.

10/7ff – If Vp/Vs-ratios greater than 3.3 indicate "saturation" (100%, I assume), how can Vp/Vs of 5 indicate only 10% saturation?

12/23 – "Largely insensitive": Not if you undershoot. "less sensitive" might be more sensitive.

12/23ff – is a discussion of the geophysical approach and could be a separate section.

13/14 – I believe it is standard in the earthquake community that you remove events before X-correlation. Could you comment on why they/you do things differently?

13/20 – I don't see how this is a conclusion. You did not use the horizontal components!

13/24 – That should go to the interpretation section!

Fig. 1: A reference to the original author of the geological map may be missing?

Fig.2: I find the colored lines/sidebars very confusing since they do not indicate profiles. It took me some time to realize that. I am not sure they are required but you could at least move them outside of the map. Also, the air photo doesn't really convey any useful information, at least none you refer to, and a simple line-drawing would do it. Perhaps the map is not even necessary at all, and Fig. 1 would suffice.

Fig 4: Please add a contour line at 1500 m/s, or adjust the color scale such that one can see this contour.

---

## Referee Comment (RC2) · Erika Angerer (Referee) · 3 Apr 2019

I agree with the summary of Prof Bleibinhaus.

The part of the paper that needs revision is the interpretation section. To me the results seem to be somewhat 'overinterpreted'. I would prefer it if this section gets shortened. The most speculative parts should be removed completely. This concerns mainly the interpretation of the resulting Vp/Vs ratio section. I find that the interpretation lacks factual evidence. Further, in Fig 11 the velocities obtained from the processing of the

active and passive seismic data are depicted next to a sonic log from a nearby well. There is a very significant difference between the seismic velocities and the sonic log velocities. This needs to be addressed in more detail.

---

## Author Comment (AC2) · 18 Apr 2019

We thank the reviewer for the comment.

The review mainly follows the line of arguments by reviewer #1, so we think most of the concerns (the interpretation section) are already addressed there. We tried to better explain the apparent differences of seismic and well log velocities by taking into account the large distance between the two sites (5 km) and a possible different properties of the alluvium / fanglomerate cover. We also provide an updated Fig. 11.

---

## Author Response (AR1)

REVIEW #1

We thank the reviewer for the detailed and very useful comments. Please find our response below.

*(1) The authors note a big difference between the two sensor types they use: 4.5-Hz 1-C-geophones of unknown making that "appear to have a better response than the ZLand 5-Hz 3-C-stations". It's unclear what exactly "better response" means (the frequency response functions are not shown) but the ZLand data in Figure 5 simply shows no useful signal. I am quite surprised by that, and I know colleagues who equipped themselves with ZLand nodes because of their supposedly great sensitivity especially at low frequencies. To claim that those instruments perform poorly has a potentially large impact on the manufacturer, and such a claim must be well founded: The authors must do more to reveal the source of the apparently poor data quality of the ZLands! Is it possible that no instrument simulation was performed? The authors don't write about it. It might explain the apparent differences. (It would also mean that the cross-correlations and the subsequent MASW would have to be re-done with the corrected data.) Or could it be a problem with the time-base?*

>> This section was indeed poorly presented. There are no issues with the nodes, but the (apparently) "better" response of the Texans is mostly due to Geology and acquisition geometry. The nodes have been deployed in the northern part, where the basement comes close to the surface and the sedimentary cover represents a presumably thick layer of heavily weathered basement rocks mixed with young soil/debris, which tend to scatter surface waves. Additionally, most of the noise sources (including the shown examples) are situated in the south, so there is an overall decay in signal strength towards the north. It is further noted that it was easier to tightly ground-couple the small 1C geophones compared to the more bulky and more heavy ZLand stations.

We chose different data examples and simply increase the gain to avoid the wrong impression of faulty node data acquisition. We also adapted the description in the text.

*(2) I find the interpretation section difficult to follow both how it is written and the conclusions it reaches. This paragraph should be rewritten for more clarity. I suggest the authors start this section summarizing all geologic information they have on the lithologies to be expected in this valley, including the geologic map and the well. More importantly, I find the lithologic description (clay/mud below ~1720 m in Figure 11) to be inconsistent with the reported P-wave velocity of 2700-3300 m/s. In their book chapter "Rock physics principles for Near Surface Geophysics" (In: SEG-Investigations in Geophysics No. 13, 2005), Knight and Enders report P-wave velocities for clay to be at most 2200 m/s. Indeed, all technical literature I am aware of specifies P-wave velocities for loose materials significantly below 2700-3200 m/s. If real, such velocity indicates lithified rocks, which might be an important finding. If not real, and if the material is indeed clay, it contradicts the statement that it is part of a 400-m-thick aquifer (13/25). Also, I find the interpretation of the GWT difficult. I fully understand that there are many reasons why $V_P$ might not be indicative for the GWT but not in this case of unconsolidated sand, where one would expect a sudden increase of $V_P$ at the GWT from maybe 800-1200 to ~1700-1800 (as seen in the well). It would be appropriate to make a first interpretation of the GWT from a contour line in the range ~1500-1800 m/s, and then check, if such a contour line coincides with the interpretation of the GWT from $V_P/V_S$-ratios.*

>> We agree that the structure of the interpretation section is poor and adds more confusion than clarity. We tried to rearrange accordingly to the remarks above.
The conclusions on water saturation, GWT interpretation, and potential aquifer properties have also been revised. In particular the comment on high Vp-velocities has been addressed more clearly, also in the context of the reviewer's comment (3) below. We point out that the interpretation of "clay" below the sand describes

the core for a few meters only, as drilling was stopped below the lacustrine sands. It has also been made more clear that the well is 5 km away, and as such cannot be used to exactly validate the results at the seismic location. Figure 11 has been updated as well.

*(3) Please add some more detail about how the reflection-processing-based Vp-velocities were derived. (At an angle of 45°, as can be seen in the reflection section, NMO velocities would be 30% increased just from the dip, not to mention the potential distortions from the Dix formula.)*

>> The NMO and interval velocities were derived from conventional CMP velocity analysis, with additional manual editing accounting for the U-shape. The extracted velocity profile is located at the center of the U-shaped structure, where reflections from the flat part of the U are actually observed. For the extracted vertical velocity profile, Vnmo velocity reductions larger than 20% for the bottom layers lead to unrealistic interval velocities (e.g. bouncing back to 1100 m/s at 250 m depth). We take this as a suggestion that the influence of the steep valley walls is not significant at this central location, but we are aware that this is a rather qualitative statement. There are also strong indications for out-of-plane reflections which might introduce non-physical layering in the velocity profile. As a result of all these uncertainties, we refrain from a detailed discussion of the lower section of the sediment fill. The text has been adapted accordingly.

*2/32 – In their model, resistivity is increased for the aquifer.*

> > Has been changed.

*3/28 – I cannot see that road on the map.*

>> This is because the narrow road in the map is masked by the signatures for receivers and shots. A description has been added to the map.

*5/9 – Over-deepening is an effect along the river-bed. How can you identify it from a cross-section?*

>> This interpretation has been expanded on in the text.

*7/28 – 8/2 Your explanation for the observation would still require that the reflections were stronger than the incident waves.*

>> This discussion has been changed/expanded.

*8/10 – "refrain" – you mean this cancels out through stacking?*

>> "refrain" has been changed to "attenuate", as stacking(averaging) with limited data will not achieve total cancellation.

*8/23 – Please clarify what you need density for. Maybe it's not so important but for unconsolidated saturated sediments, Gardner's relation tends to significantly overestimate the density.*

>> We need density as one of the model parameters (vp/vs/rho/thickness) for surface wave inversion, since the Rayleigh wave velocity is a function of Vp, Vs, and rho. Many studies have shown that the phase velocity has low sensitivity to density (e.g., Xia et al., 1999), therefore usually just constant densities like (2.0 g/cm^3) are chosen for surface wave inversion. Recent research show that the use of constant density can lead to Vs overestimation as well as create inaccurate model structures, such as a low-velocity layer (Ivanov et al., 2016). Thus, we prefer a meaningful density model which could be associated with the earth model like vp. Gardner's relation, even though it might overestimates densities, is already a significant improvement to commonly used and accepted constant densities.

*9/13 – I agree that the ratio profile length to wavelength should be at least 1.5-2. But I fail to see how you can then say that it's supposedly okay to use a ratio of less than 0.5. How does the overall length of your profile change the length of your subprofiles to which you apply MASW?*

>> As for passive MASW, there is no exact numerical relationship to indicate the maximum wavelength in relation to the linear array length. There's no clear maximum wavelength criterion, but only commonly accepted rules of thumb which will also change with the data quality, dispersion measurement, source-receiver configuration and chosen processing techniques etc. We chose the minimum frequency as 3.5 Hz due to the high-quality data and dispersion measurements (continued dispersion spectra extend as low as 2Hz) in our case. Depending on the velocity, this results in minimum wavelength-profile length factors between 0.3 and 0.7. The text has been changed accordingly.

*6/25ff (Interpretation) – Overall, I don't understand what the authors want to say in this paragraph: In the beginning the argue that there is a systematic trend regarding Vp/Vs and pore fill, and then they discuss examples that all appear to contradict those trends. Also, references to lower crustal studies, or studies where the GWT is in fractured granite, should be avoided. It's not enough for a general overview, and too much for loose sand.*

>> Despite a detailed literature search, we find very few papers which report measured Vp/Vs ratios in exploration depths corresponding to our study (e.g. below soil / weathering zone and above deep crustal targets), and in particular with relation to hydrology. Industry is expected to possess a lot of data on the reservoir level, but it is very rare that these get published. The cited studies on shallow soil structures are still considered as relevant, as they at least refer to similar material (sand, gravel). We also point out that sand represents only one among other materials (weathered granitic basement, alluvium, colluvium) which we interpret along our entire profile, so we further think that the reference to weathered granite is useful as well.
References to deep crustal studies have been eliminated.
It is not totally clear to us what the reviewer means by "[cited] examples that all appear to contradict those trends". All of the cited examples show an increase of the Vp/Vs ratio with increase of the degree of water saturation (or with the switch from dry to water-saturated materials). There was, however, a profound mis-phrasing in our description: Uyanik defines "saturation" as 100% water saturation of the pore space, and uses "water content" as the total amount of water in a volume of 100% saturated soil (defined via the weight ratio). We wrongly described the "10%-50% water content" as "10% to 50% water saturation". This error has been corrected

*10/7ff – If Vp/Vs-ratios greater than 3.3 indicate "saturation" (100%, I assume), how can Vp/Vs of 5 indicate only 10% saturation?*

>> This error has been corrected, see above.

*12/23 – "Largely insensitive": Not if you undershoot. "less sensitive" might be more sensitive.*

>> Has been changed

*12/23ff – is a discussion of the geophysical approach and could be a separate section.*

>> We reorganized the discussion / interpretation section accordingly.

*13/14 – I believe it is standard in the earthquake community that you remove events before X-correlation. Could you comment on why they/you do things differently?*

>> The events (active blasts) comprise clear and strong dispersive surface waves, which we don't want to remove. A statement has been added to section 5.1

*13/20 – I don't see how this is a conclusion. You did not use the horizontal components!*
*>> Has been removed.*

*13/24 – That should go to the interpretation section!*

>> We have changed the "vertical extent of 400 m" to "100 m", as we no longer interpret the Vp/Vs ratio below the sand. However, we think that this is a summarizing statement which fits into the conclusion section.

*Fig.2: I find the colored lines/sidebars very confusing since they do not indicate profiles. It took me some time to realize that. I am not sure they are required but you could at least move them outside of the map. Also, the air photo doesn't really convey any useful information, at least none you refer to, and a simple line-drawing would do it. Perhaps the map is not even necessary at all, and Fig. 1 would suffice.*

>> We like to keep the map as it shows the geometry of the acquisition, which is put into context into some parts of the manuscript (overall: extent of crooked line vs. 2D vertical plots; change of signal on ZLand vs. Texan recorders in relation to geology; variation of active source/signal strength along the profile has an impact of the assessment of the final velocity models). We moved the sidebars to the outside and added more useful information (elevation contours, location of the creek).

*Fig 4: Please add a contour line at 1500 m/s, or adjust the color scale such that one can see this contour.*

>> We tried to do this, but it looks odd to have an isolated contour line since the GWT is not discussed at this stage of the manuscript. However, we added Vp contour lines (1500 m/s, 1800 m/s) to the Vp/Vs plot (Fig.9) where the GWT is discussed.
* * *
REVIEW #2

We thank the reviewer for the comment. Please find our response below.

*I agree with the summary of Prof Bleibinhaus. The part of the paper that needs revision is the interpretation section. To me the results seem to be somewhat 'overinterpreted'. I would prefer it if this section gets shortened. The most speculative parts should be removed completely. This concerns mainly the interpretation of the resulting Vp/Vs ratio section. I find that the interpretation lacks factual evidence. Further, in Fig 11 the velocities obtained from the processing of the active and passive seismic data are depicted next to a sonic log from a nearby well. There is a very significant difference between the seismic velocities and the sonic log velocities. This needs to be addressed in more detail.*

>> We heavily edited the interpretation section. Due to the generality of review #2, we find it difficult to address specific points of criticism. E.g. it is not clear to us what "the lack of factual evidence" specifically addresses, e.g. if it (1) refers to a miscalculation/inaccuracy of the Vp/Vs ratio or that (2) other geological possibilities for a high Vp/Vs ratio should be discussed. In case of (1), we restricted the interpretation to the area of robust results (e.g. the very high Vp/Vs ratio in the lacustrine section. In the case of (2), our literature research indicates only saturation as a possible cause for high Vp/Vs ratios in similar geologic environments.

Regarding the difference in sonic log and seismic velocities, we point out that the locations are 5 km apart. We already discussed possible causes for difference (different composition/compaction, varying GWT) in the original manuscript at length, but we trued to improve on this description as well.

[revised manuscript text omitted]
 high P-wave velocities also lead to high $V_P/V_S$ – ratios below the sand  (3.2 – 3.8).

The interval velocities were obtained from conventional velocity analysis and the Dix equation. Steep dips as the valley flanks can lead to an overestimation of the velocities in the deeper sections of the sediment fill. However, the extracted interval velocities are located at the centre of the U-shaped valley cross section, where both reflections from the flanks and from the flat bottom do occur. Out-of-plane reflections are also present and can introduce non-physical layering in the velocity profile. Given these uncertainties, we do not attempt to correct individual stacking and interval velocities for dip but investigate the sensitivity of the $V_P/V_S$ – ratio on overall too high interval velocities in the deep section of the sediment fill. For that purpose, we reduce the P-wave velocities below the assumed bottom of the lacustrine sands by 25%. The resulting $V_P/V_S$ – ratio (dashed blue line in Fig. 11) drops to values ranging from 2.5 to 2.9. This is a large discrepancy to the uncorrected values (3.2 – 3.8), and consequently we avoid the interpretation of $V_P/V_S$ – ratio in the deep section. 
[revised manuscript text omitted]

---

## Author Response (AR2)

Comments on the revised version

Some aspects of the paper have been improved considerably but the authors reply is not nearly as detailed as my comments were, and there remain open questions, which I list again below.

(1) My question whether "*instrument simulation was performed*" remains to be answered. If the 1-Hz-geophones are typical short-period velocity sensors they undergo a 180°-phase rotation at 1 Hz. According to Ringler et al. (2018, SRL 89(5), their Fig. 5) the Z-Lands have the same type of response but undergo the phase rotation at 5 Hz. If this is not accounted for, the phase responses don't cancel in the ZL-Tx-correlations of eq 3 (as opposed to the ZL-ZL, or Tx-Tx-correlations) but are projected directly into the Green's function. Since the frequency-range 2-10 Hz was essential to the MASW, I don't understand how it was possible to successfully perform MASW with ZL-Tx-correlations. Could you provide some details on the residual reduction for the different parts of the profile? Are your segments so short that you never really use ZL-Tx-correlations?

**Response to (1a):**
**We are not sure where the reviewer got the impression from that 1 Hz geophones have been used. In fact, 4.5 Hz and 5 Hz geophones were used, which the reviewer also discussed correctly in his first revision. From the manuscript:** *"Recording stations were equipped with 385 Reftek 'Texans' data loggers / 4.5 Hz 1C geophones and with 120 Fairfield ZLand 3C 5 Hz nodes at a 5 m interval."* **4.5 Hz and 5 Hz refer to the natural frequencies of the geophones, while 1C and 3C describe the number of components. These abbreviations are conventional seismological terminology. Due to the similar natural frequencies (4.5 Hz and 5 Hz), instrument simulation was not required. This is also demonstrated by the interferogram gathers which, after taking geology into account, show identical quality of arrivals for all Virtual Source (VS) and receiver (RCV) – combinations (VS-RCV: ZLAND-ZLAND, ZLAND-TEXANS, TEXAN-ZLAND, TEXAN-TEXANS; see images below). Also, the dispersion curves of the mixed gathers are of similar quality as the dispersion curves from ZLAND-only gathers and Texans-only gathers (see additional figures below)**
**We added this explanation in the revised manuscript.**

[Figure]

**Figure above shows a comparison between virtual source gathers for different data type combinations: Left – Virtual source ZLAND, receivers to the north are LAND, recievers to the south are Texans. Right – Virtual source Texan, receivers to the north are LAND, recievers to the south are Texans.**

**The figures on the next page show dispersion curves from different intsrument combinations. Top: dispersion curve for a 100m-section comprising only ZLAND geophones; Middle: Top: dispersion curve for a 100m-section comprising ZLAND and Texan geophones; Bottom: dispersion curve for a 100m-section comprisong only Texan geophones**

[Figure]

In the revised version, you write that "*the FK transform shows that the ZLand recorders have a stronger response at low frequencies (< 5 Hz)*". In contrast, their response shows that they decay $\sim\omega^2$ below 5 Hz, where the 1-Hz-geophones are probably still on the flat passband. Please clarify.

**Response to (1b):**
**Again, given the similarity of the natural frequencies (4.5 and 5 Hz, respectively), we think that the data can be compared. However, we agree with the reviewer that the statement above is actually not accurate, as the FK transform of the entire line cannot be unambiguously separated into Texans and ZLand recorders. E.g. the apparent "stronger" response at the negative velocity branch might also be attributed to the ambient noise in the southern part of the section where Texans were deployed. As the manuscript does not aim at a detailed comparison between recording instruments, we removed this statement from the manuscript.**

(2) Please reply to my previous comment *"I fully understand that there are many reasons why $V_P$ might not be indicative for the GWT but not in this case of unconsolidated sand, where one would expect a sudden increase of $V_P$ at the GWT from maybe 800-1200 to ~1700-1800 (as seen in the well). It would be appropriate to make a first interpretation of the GWT from a contour line in the range ~1500-1800 m/s, and then check, if such a contour line coincides with the interpretation of the GWT from $V_P/V_S$-ratios."*

**Response to (2):**

**We are not sure why the reviewer has missed the consideration of this initial remark. In revision 1, we have already included a response in the manuscript, and we have already added the contour lines in the image (see below). The remark above is part of a larger general/major comment of the first review, and we have addressed it by completely re-arranging the entire interpretation section (see reply to review below, and more importantly, the entire revised manuscript with the tracked changes).**

*that it is part of a 400-m-thick aquifer (13/25). Also, I find the interpretation of the GWT difficult. I fully understand that there are many reasons why $V_P$ might not be indicative for the GWT but not in this case of unconsolidated sand, where one would expect a sudden increase of $V_P$ at the GWT from maybe 800-1200 to ~1700-1800 (as seen in the well). It would be appropriate to make a first interpretation of the GWT from a contour line in the range ~1500-1800 m/s, and then check, if such a contour line coincides with the interpretation of the GWT from $V_P/V_S$-ratios.*

>> We agree that the structure of the interpretation section is poor and adds more confusion than clarity. We tried to rearrange accordingly to the remarks above. The conclusions on water saturation, GWT interpretation, and potential aquifer properties have also been revised. In particular the comment on high Vp-velocities has been addressed more clearly, also in the context of the reviewer's comment (3) below. We point out that the interpretation of "clay" below the sand describes

**Specific consideration in the manuscript in revision 1:**

part of the profile, where West Creek occupies the lowest topographic point. The zone dips towards the north and its top is found at ca. 120 meters depth at the presumed northern edge of the over-deepened section. A northward dipping reflector is found in a comparable depth range in the seismic image, and the P-wave velocities (1500 m/s – 1800 m/s) correspond to typical velocities of saturated near-surface sands and gravels (Knights and Endres, 2005; Everett, 2013). we We therefore interpret the increased $V_P/V_S$ – ratio in the over-deepened section to represent the top of water-saturated sediments. Since the dip opposes the slope of the topography, this aquifer needs to be confined or it is leaking through fractured basement in the north. The latter hypothesis would be supported by the relatively low P- and S-wave velocities between profile distances 900 m to

**Specific consideration in the figure in the manuscript in revision 1:**

[Figure]

Figure 9: Vp/Vs ratio. White line: Interpreted top of the consolidated Precambrian basement based on P-wave refraction and reflection data. Thin black lines: Contour lines of the P-wave velocity model (Fig. 4) for 1500 m/s and 1800 m/s.

> We however added an additional description of the Vp-velocity contours vs. Vp/Vs ratio to the manuscript.

(3) Regarding the Dix-velocities and the authors reply ("*As a result of all these uncertainties, we refrain from a detailed discussion of the lower section of the sediment fill*"): Then please remove the Vp-curve and the derived Vp/Vs-curves from the Fig. 11.(Then you'd also get rid of a problem that Vp in Figure 11 indicates the bedrock at 1600 m, where it increases sharply to 3300 m/s, while you interpret it 150 m deeper, where there is no change in Vp.)

**Response to (3):**
**As discussed in the manuscript, the velocity jump to 3300 m/s at 1600 m elevation is speculated to represent a transition to Paleozoic sediments. There are three reasons why we think it is very unlikely that it represents bedrock: (1) Reflection processing shows the basement at the center of the U-shape to be at elevations between 1500 and 1400 m. (2) Velocity analysis (on which the value of 3300 m/s is based on) uses only reflections above the bedrock, and there are no indication from intra-basement reflections (e.g. velocity analysis cannot provide velocity information on the bedrock). (3) Refraction analysis (delay time model) shows bedrock velocities larger than 5500 m/s for the deepest part of the flat section of the basement, as outlined in section 4 of the manuscript.**
**We still think that the Vp-curve in the lower section is relevant for discussion, in particular in the context of possible Paleozoic sediments. The depth trend also corresponds to the Vs-curve, which taken together is supporting high velocities in the lower sediments. We agree that showing the Vp/Vs ratios of the deep section is not meaningful, and remove those Vp/Vs ratio curves from the figure.**

**In that context, the authors note that based on the seismic results they just got awarded an NSF grant to drill the deep sedimentary section. This will falsify or verify many of the interpretations expressed in this manuscript. We add that information to the manuscript in order to notify the scientific community of an upcoming "reality-check" of some of our speculations.**

(4) Regarding Gardner: Why don't you write your comment in your paper?

**Response to (4):**
**The comment has been added to the manuscript.**

(5) Regarding the ratio profile length to wavelength: Generally, the impact from being able to use 0.3 rather than 1.5 is very significant: You can use frequencies five times smaller than according to the rule-of-thumb, meaning you can look five times deeper than others. If that Is so (?), it would be important to claim it and to analyze more carefully why that is. Naively, one would expect that empirical GF from noise, because they are tainted by non-homogeneous source distribution, require tougher standards. So, what differences in source-receiver-configurations, or dispersion measurement etc, do you refer to, and what would be their impact?

**Response to (5):**

**As for the rule-of-thumb ratio number 1.5, it is regarded as a conservative threshold aims to avoid over-interpretation of the data. However, this ratio number (r) has been updated by many different studies, and we are definitely not the first ones to use values as low as 0.3. Park and Carnevale 2010 indicated that the maximum error in phase velocity is less than five percent (5%) for wavelengths (x) L ≤ x ≤ 2L, which means the optimal r could be 0.5<r<1. Pasquet et al., 2015a,b also argued that dispersion curves can be limited down to frequencies at which the spectral amplitude of the shot gather becomes too low, which indicates r could be smaller than 0.5.**

**Park, C. B., & Carnevale, M. (2010). Optimum MASW Survey — Revisit after a Decade of Use. GeoFlorida 2010: Advances in Analysis, Modeling & Design, ASCE, (Gsp 199), 1303–1312.**

**Pasquet, S., L. Bodet, A. Dhemaied, A. Mouhri, Q. Vitale, F. Rejiba, N. Flipo, and R. Guérin, 2015a, Detecting different water table levels in a shallow aquifer with combined P-, surface and SH-wave surveys: Insights from VP∕VS or Poisson's ratios: Journal of Applied Geophysics, 113,38– 50, doi: 10.1016/j.jappgeo.2014.12.005.**
**Pasquet, S., L. Bodet, L. Longuevergne, A. Dhemaied, C. Camerlynck, F. Rejiba, and R. Guérin, 2015b, 2D characterization of near-surface VP∕VS: Surface-wave dispersion inversion versus refraction tomography: Near Surface Geophysics, 13, 315–331, doi: 10.3997/1873-0604.2015028.**

**There are also many other successful real data applications with r < 1.5 for both active and passive data sets:**

**r = 0.32 Figure 10a**
**O'Connell, D. R. H., & Turner, J. P. (2011). Interferometric Multichannel Analysis of Surface Waves (IMASW). Bulletin of the Seismological Society of America, 101(No.5), 2122–2141.**

r = 0.36 Figure 9a; r = 0.15 Figure 9b
Pasquet, S., & Bodet, L. (2017). SWIP: An integrated workflow for surface-wave dispersion inversion and profiling. Geophysics, 82(6), WB47–WB61. https://doi.org/10.1190/geo2016-0625.1

r = 0.25 Figure 7
Zhang, Y., Li, Y. E., & Ku, T. (2019). Geotechnical site investigation for tunneling and underground works by advanced passive surface wave survey. Tunnelling and Underground Space Technology, 90(April), 319–329. https://doi.org/10.1016/j.tust.2019.05.003

As for the source-receiver-configurations part, I would first refer Park and Shawver 2009 work. They acquired active MASW data with different offsets and resulted in an improved depth range and imaging profiles.
Park, C. B., & Shawver, J. (2009). MASW Survey Using Multiple Source Offsets. 22nd EEGS Symposium on the Application of Geophysics to Engineering and Environmental Problems.

As for the dispersion measurement part, the poor resolution in surface wave imaging at low frequency is the key factor that affects the measured depth range and accuracy. That's why people trying to develop high resolution dispersion measurement, like Luo et al., 2007 apply high-resolution linear Radon transform for dispersion measurement;  Zheng and Hu 2017 used nonlinear signal processing technique to improve resolution.

Luo, Y., Xia, J., Miller, R. D., Xu, Y., Liu, J., & Liu, Q. (2008). Rayleigh-Wave Dispersive Energy Imaging Using a High-Resolution Linear Radon Transform. Pure and Applied Geophysics, 165(5), 903–922. https://doi.org/10.1007/s00024-008-0338-4
Zheng, Y., & Hu, H. (2017). Nonlinear signal comparison and high-resolution measurement of surface-wave dispersion. Bulletin of the Seismological Society of America, 107(3), 1551–1556. https://doi.org/10.1785/0120160242

Generally, we argue that the profile length is not the only factor that will affect the measure depth range.

The remarks above have also been integrated into the manuscript.

[revised manuscript text omitted]